# $1 + 1 < 1$? Breaking the Standalone Barrier in Federated Fine-Tuning of Multimodal Large Language Models under Non-IID Data

## Abstract

Federated fine-tuning of multimodal large language models faces significant challenges in communication costs, which can be addressed by Low-Rank Adaptation (LoRA). Existing methods typically allow all clients to collaboratively learn and share a single LoRA adapter. However, we identify a long-overlooked issue: under non-IID data, federated fine-tuning can even underperform standalone local training ("$1 + 1 < 1$"). Strikingly, much of the literature still focuses on surpassing SOTA Federated Learning (FL) methods, while neglecting the more fundamental requirement that any effective FL approach should at least outperform standalone local training. To address this, we propose a novel method termed **Fed**erated **M**ixture **o**f **L**oRA **E**xperts (Fed-MoLE). It adopts a hybrid mixture-of-LoRA-experts architecture with an alternating disentanglement–alignment mechanism. This design enables the model to disentangle diverse instance-level variations through dynamically routed LoRA experts, and then align cross-client knowledge into a unified global representation, thus enhancing robustness under non-IID data. Extensive experiments on two benchmarks show that Fed-MoLE consistently surpasses both SOAT FL baselines and standalone local training, effectively breaking the "$1 + 1 < 1$" barrier in federated fine-tuning of multimodal large language models under non-IID data.

## 1 Introduction

With the rapid development of multimodal large language models (MLLMs) (Liu et al., 2023; Driess et al., 2023), applying Federated Learning (FL) to these models has emerged as a promising paradigm for privacy-preserving training. However, traditional FL methods such as FedAvg (McMahan et al., 2017) face severe communication bottlenecks in large-scale models, as their massive parameter sizes result in substantial overhead in each communication round. To mitigate this issue, recent studies (Wu et al., 2024; Vavekanand & Sam, 2024; Ye et al., 2024b;a) have introduced Parameter-Efficient Fine-Tuning (PEFT) into FL, *e.g.*, Low-Rank Adaptation (LoRA) (Hu et al., 2022), which freezes pre-trained weights and trains only a small set of low-rank adapters, thereby reducing communication costs.

While combining FL and LoRA effectively reduces communication overhead, it faces significant challenges with real-world non-Independent and Identically Distributed (non-IID) data. In federated fine-tuning of MLLMs, client data may exhibit distribution differences, including variations in textual and visual domains (*e.g.*, medical *vs.* natural images and texts) and task types (*e.g.*, dialogue *vs.* detection). Such heterogeneity can significantly degrade the performance of federated fine-tuning. To address this, existing approaches often design refined aggregation or personalization strategies to sustain global performance under non-IID data (Sun et al., 2024; Bai et al., 2024; Long et al., 2024; Guo et al., 2025).

Despite these advances, we found a long-overlooked issue: "$1 + 1 < 1$", *i.e.*, *the performance of models trained by federated fine-tuning is even lower than models trained by* `Standalone`[1] *under non-IID data*. However, most existing studies focus primarily on surpassing FedAvg and other

---

[1]For simplicity, we refer to the method that trains models solely on local data as `Standalone`.

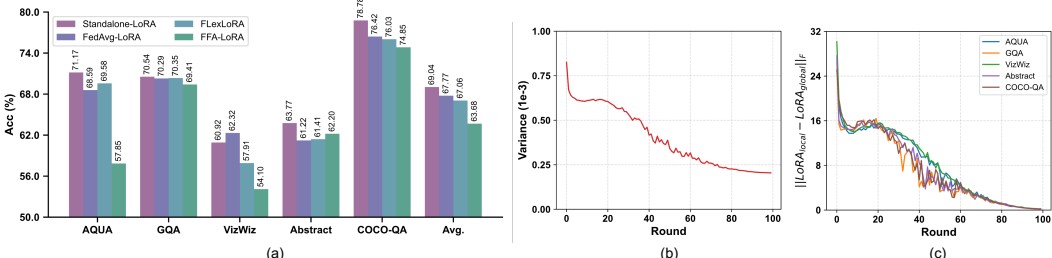

Figure 1: (a) **Accuracy (%)** of `Standalone` and various SOTA FL methods on Fed-VQA (Chen et al., 2024) benchmark with 5 clients, *i.e.*, AQUA, GQA, VizWiz, Abstract and COCO-QA, under non-IID setting. Avg. denotes the average result of all clients. (b) **Variance of local low-rank matrices** and (c) **Frobenius norm of the weight difference between local and global LoRA adapters** *versus* the communication rounds in FedAvg (McMahan et al., 2017). Higher variance indicates more pronounced client drift among local LoRA adapters, while a larger Frobenius norm reflects a greater distance between local and global LoRA adapters.

SOTA FL methods (Yan et al., 2025; Sun et al., 2024; Bai et al., 2024; Chen et al., 2024; Guo et al., 2025; Long et al., 2024), neglecting a more fundamental principle: *any effective FL method should at least outperform `Standalone`; otherwise, collaborative training becomes meaningless*. As shown in Figure 1 (a), the results of `Standalone` on the testing datasets of the majority of clients, as well as the average result, are higher than those of various FL methods. This indicates traditional FL not only fails to deliver the expected benefits but may even degrade performance. This is because learning a single global low-rank adapter fails to capture the distribution differences across clients. The client drift (Karimireddy et al., 2020; Li et al., 2020) in local low-rank adapters (see Figure 1 (b)) caused by non-IID data leads to a representation mismatch (see Figure 1 (c)) between the aggregated global low-rank adapter and the local data, thereby significantly degrading performance.

The Mixture-of-Experts (MoE) (Shazeer et al., 2017; Lepikhin et al., 2020; Chen et al., 2023; Ma et al., 2018; Zhang et al., 2024b) employs sparse activation with adaptive routing to activate only a few experts per input, efficiently modeling diverse data distributions and demonstrating strong potential across domains and tasks. Inspired by this, we propose Federated Mixture of LoRA Experts (Fed-MoLE) to break the "$1 + 1 < 1$" barrier in federated fine-tuning under non-IID conditions. Fed-MoLE adopts a hybrid mixture-of-LoRA-experts architecture with an alternating disentanglement–alignment mechanism: dynamically routed LoRA experts capture instance-level variations, while a shared LoRA expert integrates cross-client knowledge into a robust global representation. This design enables the model to benefit from collaboration while effectively handling heterogeneous data distributions, achieving "$1 + 1 > 1$" without introducing extra communication cost. Experiments on two benchmarks show that Fed-MoLE consistently outperforms both `Standalone` and SOTA FL baselines, validating its effectiveness under non-IID data.

- We identify the overlooked problem of federated fine-tuning of MLLM under non-IID data, where FL can perform even worse than `Standalone` ("$1 + 1 < 1$"), challenging the fundamental necessity of collaboration.

- We propose Fed-MoLE, a hybrid mixture-of-LoRA-experts framework with an alternating disentanglement–alignment mechanism, enabling instance-adaptive personalization and robust cross-client knowledge integration to surpass standalone local training.

- Through extensive experiments on two benchmarks, Fed-MoLE consistently surpasses both SOTA FL baselines and `Standalone`, demonstrating its effectiveness in breaking the standalone barrier under non-IID data.

## 2 RELATED WORK

**Multimodal Large Language Models.** MLLMs, such as LLaVA (Liu et al., 2023), PaLM-E (Driess et al., 2023), Qwen2-VL (Wang et al., 2024a) and GPT-4V (Achiam et al., 2023), have demonstrated remarkable capabilities in multimodal understanding and generation. These models are pre-trained on massive image-text pairs collected from web-scale datasets, enabling effective alignment of visual

and linguistic representations. Their cross-modal versatility has facilitated widespread adoption in downstream tasks such as Visual Question Answering (VQA) (Wang et al., 2022; Shao et al., 2023), image captioning (Luo et al., 2024; Xie et al., 2022), and visual reasoning (Dong et al., 2025; Schulze Buschoff et al., 2025). To further tailor these models to specific applications, fine-tuning techniques such as visual instruction tuning (Liu et al., 2023) have been proposed, enabling practical deployment of MLLMs (Li et al., 2023; Yuan et al., 2024; Li et al., 2024). However, these approaches rely heavily on centralized data collection and training. In practice, visual and textual data are often distributed across users' personal devices (*e.g.*, smartphones and AR glasses), making centralized training impractical due to privacy concerns. In this work, we aim to address this challenge by enabling decentralized adaptation of MLLMs while preserving user privacy.

**Parameter-Efficient Fine-Tuning.** Traditional full-model fine-tuning incurs substantial training overhead when applied to large models. PEFT (Mangrulkar et al., 2022; Xu et al., 2023) has emerged as a popular solution for adapting large pre-trained models to downstream tasks with fewer trainable parameters. Instead of updating the entire model, PEFT modifies only a small set of parameters, such as those at specific layers (Houlsby et al., 2019; Mou et al., 2024) or input positions (Li & Liang, 2021; Lester et al., 2021; Jia et al., 2022), while keeping most weights fixed. Among these methods, LoRA (Hu et al., 2022) has gained significant attention for its simplicity and effectiveness. By injecting a pair of low-rank matrices into selected layers, LoRA significantly reduces training costs while maintaining strong downstream performance. As a result, LoRA has become a mainstream approach for efficient fine-tuning of large models, prompting continued efforts to refine its design for improved adaptability and performance (Liu et al., 2024; Zhang et al., 2023; Gao et al., 2024; Lin et al., 2024; Meng et al., 2024; Si et al., 2024). Building on this, recent work (Wu et al., 2024; Vavekanand & Sam, 2024; Ye et al., 2024b;a) has explored integrating LoRA with FL to reduce communication costs during distributed fine-tuning. Moreover, some studies (Wang et al., 2024b; Sun et al., 2024; Bai et al., 2024; Yan et al., 2025) have attempted to improve the performance of FL combined with LoRA under non-IID settings. However, these works mainly focus on outperforming other FL methods, while overlooking the fundamental effectiveness issue of FL under non-IID data. In this work, we reveal an important issue of federated fine-tuning of MLLMS under non-IID data that its performance even lower than `Standalone`.

**Federated Learning with Non-IID Data.** FedAvg (McMahan et al., 2017), a pioneering algorithm, established the standard FL paradigm by training local models on individual clients and updating a global model on the server through parameter averaging. Its simple yet effective design has facilitated widespread adoption across a variety of applications, particularly in privacy-sensitive domains such as finance (Long et al., 2020; Chatterjee et al., 2023) and healthcare (Feng et al., 2022; Yan et al., 2024; Jiang et al., 2023a). However, its performance deteriorates significantly under non-IID data due to client drift (Karimireddy et al., 2020), making handling non-IID data effectively a fundamental challenge in FL. Among the various types of non-IID data, domain shift (Tan et al., 2022; T Dinh et al., 2020; Arivazhagan et al., 2019; Jiang et al., 2023b; Wang et al., 2023; Fallah et al., 2020) has attracted considerable attention, where each client's data originates from different domains. This setting is similar to the non-IID scenarios in fine-tuning multi-modal large language models. However, most prior works train full models from scratch, which is not directly applicable to federated fine-tuning of MLLMs, as LoRA-based adaptation differs fundamentally from full-model training. Furthermore, existing studies typically address non-IID challenges in a single modality (e.g., images or text), whereas federated fine-tuning of MLLMs involves heterogeneity across modalities and tasks, making the problem more challenging. To address this issue, this paper proposes a LoRA-based FL approach to enable effective federated fine-tuning of MLLMs.

## 3 METHODOLOGY

### 3.1 PRELIMINARY

**Low-Rank Adaptation.** LoRA offers an efficient fine-tuning strategy to adapt MLLMs to downstream tasks. Instead of fine-tuning the entire model, LoRA selectively updates parameters at specific locations, such as the self-attention layers. Specifically, given a pre-trained weight matrix $\boldsymbol{W}^0 \in \mathbb{R}^{d_1 \times d_2}$ from a target layer, LoRA keeps it frozen and introduces two trainable low-rank matrices, $\boldsymbol{B} \in \mathbb{R}^{d_1 \times r}$ and $\boldsymbol{A} \in \mathbb{R}^{r \times d_2}$, to model the weight update $\Delta \boldsymbol{W}$, where rank $r \ll \min(d_1, d_2)$.

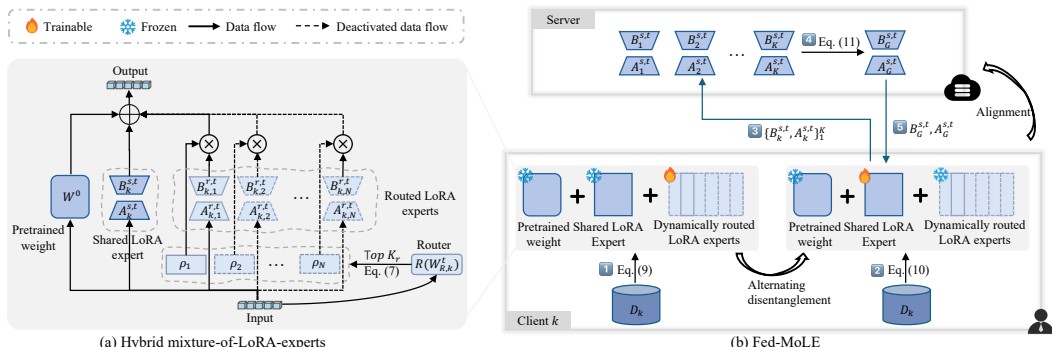

(a) Hybrid mixture-of-LoRA-experts  (b) Fed-MoLE

Figure 2: **Illustration of Fed-MoLE**. (a) depicts the architecture of the proposed hybrid Mixture-of-LoRA-Experts, and (b) presents the overall training pipeline. It employs an alternating disentanglement–alignment mechanism to disentangle diverse instance-level variations through dynamically routed LoRA experts, and then align cross-client knowledge into a unified global representation.

After training, the updated weight $\hat{W}$ can be written as:

$$\hat{W} = W^0 + \Delta W = W^0 + BA. \tag{1}$$

To align the pre-trained weight at the beginning, $B$ is typically initialized to zeros, while $A$ is randomly initialized using a normal distribution $\mathcal{N}(0, \sigma^2)$. Additionally, a scaling coefficient $\alpha$ is typically introduced to control the magnitude of the low-rank update, applied as a factor of $\alpha/r$.

**Federated Learning with LoRA.** Consider $K$ clients, where each client $k \in [K]$ holds a local dataset $D_k$ for the downstream task, *e.g.*, VQA, consisting of $n_k$ training samples $(x_i, y_i)_{i=1}^{n_k}$, drawn from a joint distribution $P_k(x, y)$. The objective is to fine-tune a MLLM $f(W^0)$ on these distributed datasets. Due to the substantial number of parameters in MLLMs, a feasible solution is to combine FL with LoRA. Specifically, each client introduces trainable low-rank matrices and optimizes them locally by minimizing the empirical risk over its own dataset. The global objective $\mathcal{L}_G$ and the local objective $\mathcal{L}_k$ at communication round $t \in [T]$ are defined as:

$$\mathcal{L}_G = \sum_{k=1}^{K} \gamma_k \mathcal{L}_k, \quad \text{and} \tag{2}$$

$$\mathcal{L}_k = \frac{1}{n_k} \sum_{(x_i, y_i) \sim D_k} \ell(f(x_i; W^0; B_k^t; A_k^t); y_i), \tag{3}$$

where $\gamma_k = \frac{n_k}{\sum_{i=1}^{K} n_i}$, and $\ell(\cdot)$ denotes the loss function. $B_k^t$ and $A_k^t$ are the local low-rank matrices, initialized from the global matrices of the previous round, $B_G^{t-1}$ and $A_G^{t-1}$, where $B_G^0 = \mathbf{0}$ and $A_G^0 \sim \mathcal{N}(0, \sigma^2)$. The local update step can be described as:

$$B_k^t \leftarrow B_k^t - \eta \nabla \mathcal{L}_k, \quad A_k^t \leftarrow A_k^t - \eta \nabla \mathcal{L}_k, \tag{4}$$

where $\eta$ is the learning rate. After local training, the updated low-rank matrices are sent to the server for aggregation:

$$B_G^t = \sum_{k=1}^{K} \gamma_k B_k^t, \quad A_G^t = \sum_{k=1}^{K} \gamma_k A_k^t. \tag{5}$$

The aggregated global matrices $B_G^t$ and $A_G^t$ are then distributed to all clients as initialization for the next communication round.

## 3.2 FEDERATED MIXTURE OF LORA EXPERTS

Under the non-IID setting, clients have data drawn from different distributions $P_k$, which induces substantial client drift in local LoRA adapters. As a result, the global LoRA adapter $B_G^t$ and $A_G^t$ deviates significantly from the local LoRA adapter $B_k^t$ and $A_k^t$, resulting in representation mismatch between the global LoRA adapter and local data. In this work, we found that the above naive

---

**Algorithm 1:** `Fed-MoLE`

---

**Input:** Number of clients $K$, communication rounds $T$, learning rate $\eta$, Pre-trained weight $\boldsymbol{W}^0$, Datasets $D_1, D_2, \ldots, D_K$

**Output:** $K$ personalized models

1 Initialize $\{\boldsymbol{B}_G^{s,0}; \boldsymbol{A}_G^{s,0}\}$ and $\{\{\boldsymbol{B}_{k,j}^{r,0}\}_{j=1}^N; \{\boldsymbol{A}_{k,j}^{r,0}\}_{j=1}^N; \boldsymbol{W}_{\mathcal{R},k}^0\}_{k=1}^K$

2 **for** *round* $t = 1, 2, \ldots, T$ **do**

3    **for** *client* $k = 1, 2, \ldots, K$ ***parallelly*** **do**

4      $\boldsymbol{B}_k^{s,t} \leftarrow \boldsymbol{B}_G^{s,t-1}, \quad \boldsymbol{A}_k^{s,t} \leftarrow \boldsymbol{A}_G^{s,t-1}, \quad \boldsymbol{W}_{\mathcal{R},k}^t \leftarrow \boldsymbol{W}_{\mathcal{R},k}^{t-1}$

5      $\forall j \in [N], \quad \boldsymbol{B}_{k,j}^{r,t} \leftarrow \boldsymbol{B}_{k,j}^{r,t-1}, \quad \boldsymbol{A}_{k,j}^{r,t} \leftarrow \boldsymbol{A}_{k,j}^{r,t-1}$

6      **for** $(x_i, y_i) \sim D_k$ **do**

7        $\mathcal{L}_k \leftarrow \ell(\boldsymbol{H}(x_i; \boldsymbol{W}^0; \boldsymbol{B}_k^{s,t}; \boldsymbol{A}_k^{s,t}; \{\boldsymbol{B}_{k,j}^{r,t}\}_{j=1}^N; \{\boldsymbol{A}_{k,j}^{r,t}\}_{j=1}^N; \boldsymbol{W}_{\mathcal{R},k}^t); y_i)$

8        $\{\{\boldsymbol{B}_{k,j}^{r,t}\}_{j=1}^N; \{\boldsymbol{A}_{k,j}^{r,t}\}_{j=1}^N; \boldsymbol{W}_{\mathcal{R},k}^t\} \leftarrow \{\{\boldsymbol{B}_{k,j}^{r,t}\}_{j=1}^N; \{\boldsymbol{A}_{k,j}^{r,t}\}_{j=1}^N; \boldsymbol{W}_{\mathcal{R},k}^t\} - \eta \nabla \mathcal{L}_k$

9      **end**

10      **for** $(x_i, y_i) \sim D_k$ **do**

11        $\mathcal{L}_k \leftarrow \ell(\boldsymbol{H}(x_i; \boldsymbol{W}^0; \boldsymbol{B}_k^{s,t}; \boldsymbol{A}_k^{s,t}; \{\boldsymbol{B}_{k,j}^{r,t}\}_{j=1}^N; \{\boldsymbol{A}_{k,j}^{r,t}\}_{j=1}^N; \boldsymbol{W}_{\mathcal{R},k}^t); y_i)$

12        $\{\boldsymbol{B}_k^{s,t}; \boldsymbol{A}_k^{s,t}\} \leftarrow \{\boldsymbol{B}_k^{s,t}; \boldsymbol{A}_k^{s,t}\} - \eta \nabla \mathcal{L}_k$

13      **end**

14    **end**

15    $\boldsymbol{B}_G^{s,t} \leftarrow \sum_{k=1}^K \gamma_k \boldsymbol{B}_k^{s,t}, \quad \boldsymbol{A}_G^{s,t} \leftarrow \sum_{k=1}^K \gamma_k \boldsymbol{A}_k^{s,t}$

16 **end**

17 **return** $\{\boldsymbol{B}_G^{s,T}; \boldsymbol{A}_G^{s,T}; \{\boldsymbol{B}_{k,j}^{r,T}\}_{j=1}^N; \{\boldsymbol{A}_{k,j}^{r,T}\}_{j=1}^N; \boldsymbol{W}_{\mathcal{R},k}^T\}_{k=1}^K$

---

solution can even underperform `Standalone`, *i.e.*, "$1 + 1 < 1$" issue, highlighting challenges for the real-world deployment of federated fine-tuning of MLLMs. To address this, we propose a novel Federated Mixture of LoRA Experts (Fed-MoLE), illustrated in Figure 2. It comprises two key components: a hybrid mixture-of-LoRA-experts architecture and an alternating disentanglement–alignment mechanism.

**Hybrid Mixture-of-LoRA-Experts.** Figure 2 (a) illustrates the basic architecture of our proposed hybrid mixture of LoRA experts. Instead of learning a single LoRA adapter, each client introduces multiple parallel LoRA experts[2] for each target layer of the base model $\boldsymbol{W}^0 \in \mathbb{R}^{d_1 \times d_2}$, including one shared LoRA expert $(\boldsymbol{B}_k^{s,t}, \boldsymbol{A}_k^{s,t})$ and $N$ routed LoRA experts $\{(\boldsymbol{B}_{k,i}^{r,t}, \boldsymbol{A}_{k,i}^{r,t})\}_{i=1}^N$, which are activated by an instance-adaptive router $\mathcal{R}(\boldsymbol{W}_{\mathcal{R},k}^t)$. The router dynamically selects a subset of routed LoRA experts for each instance, enabling the model to capture instance-level variations. Since the shared LoRA expert is activated for all data samples, it can effectively learn the general patterns shared across different data samples.

Given the input $\boldsymbol{X}$ (*i.e.*, the output of the previous layer), the output $\boldsymbol{H}(\boldsymbol{X})$ of our hybrid mixture-of-LoRA-experts is defined as:

$$\boldsymbol{H}(\boldsymbol{X}) = f(\boldsymbol{X}; \boldsymbol{W}^0) + \boldsymbol{B}_k^{s,t} \boldsymbol{A}_k^{s,t} \boldsymbol{X} + \sum_{i=1}^N \rho_i \boldsymbol{B}_{k,i}^{r,t} \boldsymbol{A}_{k,i}^{r,t} \boldsymbol{X}, \tag{6}$$

where $\rho = [\ldots, \rho_i, \ldots] \in \mathbb{R}^N$ denotes the output scores of the router. We apply a $TopK_r(\cdot)$ strategy to select the top $K_r$ scores and activate only the routed LoRA experts corresponding to the selected paths. This can be written as:

$$\rho_i = \frac{\hat{z}_i}{\sum_{j=1}^N \hat{z}_j}, \quad \hat{z}_i = \begin{cases} z_i, & z_i \in TopK_r(z), \\ 0, & otherwise, \end{cases} \quad z = \mathcal{R}(\boldsymbol{X}; \boldsymbol{W}_{\mathcal{R},k}^t). \tag{7}$$

It is worth noting that, since only a small subset of routed LoRA experts is activated, the remaining experts do not participate in subsequent computations. This sparse structure significantly reduces the additional computational overhead when $N$ is larger.

---

[2] We define a complete LoRA adapter as an expert.

**Alternating Disentanglement–Alignment.** With the proposed hybrid mixture-of-LoRA-experts, for client $k$ at round $t$, the local objective in Eq. (3) can be rewritten as follows:

$$\mathcal{L}_k = \frac{1}{n_k} \sum_{(x_i,y_i)\sim D_k} \ell(\boldsymbol{H}(x_i; \boldsymbol{W}^0; \boldsymbol{B}_k^{s,t}; \boldsymbol{A}_k^{s,t}; \{\boldsymbol{B}_{k,j}^{r,t}\}_{j=1}^N; \{\boldsymbol{A}_{k,j}^{r,t}\}_{j=1}^N; \boldsymbol{W}_{\mathcal{R},k}^t\}; y_i). \quad (8)$$

During training, if we directly optimize Eq. (8) to update and share all trainable components across clients, the shared and routed experts would interfere with each other, resulting in entangled knowledge and suboptimal performance. Meanwhile, the fine-grained instance-level differences captured by the routed LoRA experts are diminished during global aggregation. To address this, we introduce an alternating disentanglement–alignment mechanism to disentangle diverse instance-level variations and align cross-client knowledge into a unified global representation. Specifically, we divide the procedure of local training into two phases, as shown in Figure 2 (b). In the first phase, the shared LoRA expert is frozen while only the routed LoRA experts and the router are updated. The update procedure is given by:

$$\{\{\boldsymbol{B}_{k,j}^{r,t}\}_{j=1}^N; \{\boldsymbol{A}_{k,j}^{r,t}\}_{j=1}^N; \boldsymbol{W}_{\mathcal{R},k}^t\} \leftarrow \{\{\boldsymbol{B}_{k,j}^{r,t}\}_{j=1}^N; \{\boldsymbol{A}_{k,j}^{r,t}\}_{j=1}^N; \boldsymbol{W}_{\mathcal{R},k}^t\} - \eta \nabla \mathcal{L}_k. \quad (9)$$

In the next phase, we freeze the updated routed LoRA experts and router, and update the shared LoRA expert as follows:

$$\{\boldsymbol{B}_k^{s,t}; \boldsymbol{A}_k^{s,t}\} \leftarrow \{\boldsymbol{B}_k^{s,t}; \boldsymbol{A}_k^{s,t}\} - \eta \nabla \mathcal{L}_k. \quad (10)$$

The above disentanglement reduces gradient conflicts between the shared and routed LoRA experts, encouraging the shared expert to capture common patterns across samples while enabling the routed experts to model instance-level differences. After local training, the parameters of the shared expert are transmitted to the server for averaging, thereby aligning general knowledge across clients into a unified global representation:

$$\boldsymbol{B}_G^{s,t} = \sum_{k=1}^K \gamma_k \boldsymbol{B}_k^{s,t}, \quad \boldsymbol{A}_G^{s,t} = \sum_{k=1}^K \gamma_k \boldsymbol{A}_k^{s,t}. \quad (11)$$

Notably, the shared LoRA expert focuses more on common patterns across samples by decoupling. This brings the shared LoRA experts across clients closer together, thereby reducing client drift and enhancing the global representation. Finally, $\boldsymbol{B}_G^{s,t}$ and $\boldsymbol{A}_G^{s,t}$ are sent back to the clients to initialize each client's shared expert and start the next training round.

## 4 EXPERIMENTS

### 4.1 EXPERIMENTAL SETUP

**Datasets.** We conducted experiments on two federated fine-tuning benchmarks, including **Fed-VQA** and **Fed-Med**. Fed-VQA (Chen et al., 2024) comprises five different VQA datasets: AQUA (Garcia et al., 2020), GQA (Hudson & Manning, 2019), VizWiz (Gurari et al., 2018), Abstract (Antol et al., 2015), and COCO-QA (Ren et al., 2015). Each dataset is treated as an individual client. The datasets exhibit diverse visual and textual characteristics, representing a typical non-IID scenario across different clients. In contrast to Fed-VQA, where all clients are aligned to the VQA task, Fed-Med (Zheng et al., 2025) comprises three clients with data drawn from different tasks, including VQA, report generation and detection, arising task-level heterogeneity. The data of Fed-Med is from SLAKE (Liu et al., 2021) (VQA), VQA-RAD (Lau et al., 2018) (VQA), MIMIC-CXR (Johnson et al., 2019) (report generation) and RadGenome-Chest CT (Zhang et al., 2024a) (detection) datasets. Each client in both Fed-VQA and Fed-Med has a training set and a test set. We perform federated fine-tuning using the training sets of all clients and evaluate the fine-tuned model on the test set of each client. Since answers in the VQA task are generally single-word responses, we adopt accuracy (%) as the evaluation metric, counting a prediction as correct only when it exactly matches the ground-truth answer. For report generation, we employ Qwen3-8B (Yang et al., 2025) as a judge to evaluate the quality of reports generated by the MLLMs against the ground-truth reports, yielding a final score (%) on a scale from 100% (poor) to 500% (excellent). Besides, we use Intersection over Union (IoU (%)) as the evaluation metric for the detection task.

**Baselines.** We compare our method with `Standalone`, where each client performs only local training, as well as nine SOTA global model learning approaches, including **FedAvg** (McMahan

Table 1: **Quantitative comparison of all methods** on **Fed-VQA** (Chen et al., 2024) benchmark comprising five clients: AQUA (Garcia et al., 2020), GQA (Hudson & Manning, 2019), VizWiz (Gurari et al., 2018), Abstract (Antol et al., 2015), and COCO-QA (Ren et al., 2015). We report the test accuracy (%) of each method on every client's dataset, along with the average (Avg.) across all clients. The best results are marked in **bold**.

| Method | AQUA Acc(%) | Δ | GQA Acc(%) | Δ | VizWiz Acc(%) | Δ | Abstract Acc(%) | Δ | COCO-QA Acc(%) | Δ | Avg.(%) | Δ |
|---|---|---|---|---|---|---|---|---|---|---|---|---|
| Standalone | 71.17 | - | 70.54 | - | 60.92 | - | 63.77 | - | 78.78 | - | 69.04 | - |
| *Global Model Learning* | | | | | | | | | | | | |
| FedAvg | 68.59 | 2.58↓ | 70.29 | 0.25↓ | 62.32 | 1.40↑ | 61.22 | 2.55↓ | 76.42 | 2.36↓ | 67.77 | 1.27↓ |
| FedProx | 67.99 | 3.18↓ | **71.66** | 1.12↑ | 58.91 | 2.01↓ | 63.58 | 0.19↓ | 76.81 | 1.97↓ | 67.79 | 1.25↓ |
| FedAdam | 70.17 | 1.00↓ | 68.29 | 2.25↓ | 56.51 | 4.41↓ | 61.81 | 1.96↓ | 78.19 | 0.59↓ | 66.99 | 2.05↓ |
| FedYogi | 68.78 | 2.39↓ | 70.16 | 0.38↓ | 48.49 | 12.43↓ | 62.40 | 1.37↓ | 78.97 | 0.19↑ | 65.76 | 3.28↓ |
| FedAdagrad | 67.79 | 3.38↓ | 70.54 | 0.00↓ | 56.51 | 4.41↓ | 62.40 | 1.37↓ | 77.60 | 1.18↓ | 66.97 | 2.07↓ |
| Scaffold | 69.18 | 1.99↓ | 70.91 | 0.37↑ | 56.91 | 4.01↓ | 60.82 | 2.95↓ | 77.40 | 1.38↓ | 67.05 | 1.99↓ |
| FFA-LoRA | 57.85 | 13.32↓ | 69.41 | 1.13↓ | 54.10 | 6.82↓ | 62.20 | 1.57↓ | 74.85 | 3.93↓ | 63.68 | 5.36↓ |
| FlexLoRA | 69.58 | 1.59↓ | 70.35 | 0.19↓ | 57.91 | 3.01↓ | 61.41 | 2.36↓ | 76.03 | 2.75↓ | 67.06 | 1.98↓ |
| FRLoRA | 72.56 | 1.39↑ | 70.35 | 0.19↓ | 60.52 | 0.40↓ | **65.74** | 1.97↑ | 74.45 | 4.33↓ | 68.72 | 0.32↓ |
| *Personalized Model Learning* | | | | | | | | | | | | |
| FedDPA-LoRA | 72.76 | 1.59↑ | 70.16 | 0.38↓ | 60.12 | 0.80↓ | 65.74 | 1.97↑ | 78.97 | 0.19↑ | 69.55 | 0.51↑ |
| FedSA-LoRA | 70.97 | 0.20↓ | 69.98 | 0.56↓ | **63.72** | 2.80↑ | 64.56 | 0.79↑ | **79.96** | 1.18↑ | 69.84 | 0.80↑ |
| FedLEASE | 70.77 | 0.40↓ | 70.31 | 0.19↓ | 63.12 | 2.20↑ | 66.53 | 2.76↑ | 77.40 | 1.38↓ | 69.71 | 0.67↑ |
| **Fed-MoLE** | **74.95** | 3.78↑ | 71.10 | 0.56↑ | 61.32 | 0.40↑ | 64.96 | 1.19↑ | 79.37 | 0.78↑ | **70.38** | 1.34↑ |

et al., 2017), **FedProx** (Li et al., 2020), **Scaffold** (Karimireddy et al., 2020), **FedAdagard** (Reddi et al., 2021), **FedYogi** (Reddi et al., 2021), **FedAdam** (Reddi et al., 2021), **FlexLoRA** (Bai et al., 2024), **FFA-LoRA** (Sun et al., 2024), and **FRLoRA** (Yan et al., 2025), all of which aim to learn a single global LoRA adapter shared across clients. In addition, we compare against two personalized model learning approaches based on LoRA, **FedDPA-LoRA** (Long et al., 2024), **FedLEASE** (Wang et al., 2025) and **FedSA-LoRA** (Guo et al., 2025), which learns a personalized model for each client.

**Implementation Details.** We utilize **LLaVA-1.5-7B** (Liu et al., 2023) as the base model, with pretrained weights obtained from Hugging Face. Local training is performed via visual instruction tuning (Liu et al., 2023), following the instruction template as:

```
USER: <image>
{question} ASSISTANT:
```

<image> denotes the image token, and {question} represents the question content. For LoRA, we set the rank parameter $r = 8$, the scaling factor $\alpha = 16$, and apply a dropout rate of 0.05. In Fed-VQA, LoRA adapters are integrated into the q_proj, k_proj, v_proj, and up_proj layers of the base model, whereas in Fed-Med, they are applied only to the up_proj and down_proj layers. Moreover, Fed-MoLE is applied only to the up_proj layer, while the other layers use the standard LoRA implementation. We optimize the models using the AdamW optimizer with a batch size of 8 for Fed-VQA and 4 for Fed-Med. For both benchmarks, we use an initial learning rate of 5e-5 with a cosine annealing learning rate schedule. All methods are trained for 100 communication rounds, with 50 local update steps per round. All methods are implemented in PyTorch and evaluated on four NVIDIA RTX 4090 GPUs with 24 GB memory each. The number of routed LoRA experts $N$ and hyper-parameter $K_r$ of the Top$K_r$ selection strategy in our method are set to 8 and 2 by default.

## 4.2 MAIN RESULTS

Table 1 and Table 2 present the results of all methods on the Fed-VQA and Fed-Med benchmarks, respectively. To provide a comprehensive evaluation, we report the evaluation metrics on each client's test set and the average (Avg.) across all clients. The evaluation on each client's test set reflects local performance, while the average across clients serves as an indicator of global performance.

Table 2: **Quantitative comparison of all methods** on **Fed-Med** (Zheng et al., 2025) benchmark comprising three clients with different tasks: VQA, report generation and detection. We report the evaluation metrics (%) for each method on every client's dataset, along with the average (Avg.) across all clients. The best results are marked in **bold**.

| Method | VQA Acc(%) | Δ | Report Generation Score(%) | Δ | Detection IoU(%) | Δ | Avg.(%) | Δ |
|---|---|---|---|---|---|---|---|---|
| Standalone | 65.50 | - | 224.40 | - | 40.62 | - | 110.17 | - |
| *Global Model Learning* | | | | | | | | |
| FedAvg | 62.00 | 3.50 ↓ | 219.76 | 4.64 ↓ | 39.29 | 1.33 ↓ | 107.01 | 3.16 ↓ |
| FedProx | 63.00 | 2.50 ↓ | 179.88 | 44.52 ↓ | 39.31 | 1.31 ↓ | 94.06 | 16.11 ↓ |
| FedAdam | 65.00 | 0.50 ↓ | 221.32 | 3.08 ↓ | 39.56 | 1.06 ↓ | 108.62 | 1.55 ↓ |
| FedYogi | 63.50 | 2.00 ↓ | 212.88 | 11.52 ↓ | 38.40 | 2.22 ↓ | 104.92 | 5.25 ↓ |
| FedAdagrad | 63.00 | 3.50 ↓ | 190.19 | 34.21 ↓ | 37.46 | 3.16 ↓ | 96.88 | 13.29 ↓ |
| Scaffold | 64.00 | 1.50 ↓ | 186.45 | 37.95 ↓ | 38.32 | 2.30 ↓ | 96.25 | 13.92 ↓ |
| FFA-LoRA | 65.00 | 0.50 ↓ | 171.85 | 52.55 ↓ | 36.29 | 4.33 ↓ | 91.04 | 19.13 ↓ |
| FlexLoRA | 65.50 | 0.00 ↓ | 207.81 | 16.59 ↓ | 36.54 | 4.08 ↓ | 106.61 | 6.88 ↓ |
| FRLoRA | 66.00 | 0.50 ↑ | 218.19 | 6.21 ↓ | 42.62 | 2.00 ↑ | 108.93 | 1.23 ↓ |
| *Personalized Model Learning* | | | | | | | | |
| FedDPA-LoRA | 66.50 | 1.00 ↑ | 230.75 | 6.35 ↑ | 37.03 | 3.59 ↓ | 111.42 | 1.25 ↑ |
| FedSA-LoRA | 67.00 | 1.50 ↑ | 234.80 | 10.30 ↑ | 39.86 | 19.13 ↓ | 113.88 | 3.71 ↑ |
| FedLEASE | 64.50 | 1.00 ↓ | 239.71 | 15.31 ↓ | 43.37 | 2.75 ↑ | 115.86 | 5.69 ↑ |
| **Fed-MoLE** | **68.00** | 2.50 ↑ | **260.00** | 35.60 ↑ | **46.38** | 5.76 ↑ | **124.79** | 14.62 ↑ |

On both Fed-VQA and Fed-Med, all global model learning methods suffer performance degradation compared to Standalone. For instance, the global performance of FFA-LoRA decreases by **5.36%** and **19.13%**, respectively, and yields lower accuracy on all clients. This clearly demonstrates that learning a single global LoRA adapter cannot effectively handle the diverse data distributions across clients, resulting in the "$1 + 1 < 1$" issue. In contrast, on both benchmarks, Fed-MoLE consistently outperforms the local performance of Standalone across all clients and further improves the global performance, raising it from **69.04%** to **70.38%** on Fed-VQA and from **110.17%** to **124.79%** on Fed-Med. This validates the effectiveness of our method, which enables the model to leverage collaboration while robustly handling heterogeneous data distributions, thereby breaking the Standalone barrier.

Besides, Fed-MoLE consistently outperforms other SOTA FL methods on both benchmarks, *e.g.*, it improves global performance over FedAvg by **3.61%** on Fed-VQA and **17.78%** on Fed-Med. Even compared with advanced personalized approaches, Fed-MoLE achieves further improvements, surpassing FedSA-LoRA by **0.54%** on Fed-VQA and **10.71%** on Fed-Med, as these personalized methods focus on adapting to local distributions while ignoring fine-grained instance-level variations. The above results highlight the superiority of Fed-MoLE in handling distribution difference across clients in text, image and task.

### 4.3 ANALYTICAL STUDIES

**Ablation Study.** To investigate the effectiveness of the two key modules in our method, *i.e.*, Hybrid MoE (HMoE) and Alternating Disentanglement–Alignment (ADA) mechanism, we construct a variant, Fed-MoLE-V1, which corresponds to our method without the ADA module. When both key modules are removed, the method degenerates to standard FedAvg+LoRA. The results of our ablation study are presented in Table 3. As shown, both modules contribute to significant performance improvements, facilitating more effective handling of data heterogeneity across clients.

**Architecture Design.** To deeply investigate the impact of different modules in our hybrid MoE, *i.e.*, the shared LoRA expert, routed LoRA experts and the instance-adaptive router, we construct three variants of Fed-MoLE: ❶ M1: Align all modules across clients, rather than only the shared expert. ❷ M2: Train the entire hybrid MoE locally without aggregating any local knowledge. ❸ M3: In addition to the shared expert, also align instance-level routers from all clients to reach a unified global decision for each data sample. As shown in Table 4, compared to our method, both M1 and M3

Table 3: **Ablation study** of Fed-MoLE on Fed-VQA.

| Method | HMoE | ADA | Avg. | Δ |
|---|---|---|---|---|
| FedAvg | ✗ | ✗ | 67.77 | 2.61 ↓ |
| Fed-MoLE-V1 | ✓ | ✗ | 69.40 | 0.98 ↓ |
| **Fed-MoLE** | ✓ | ✓ | **70.38** | - |

Table 4: **Results** of different architectures on Fed-VQA.

| Method | Avg. | Δ |
|---|---|---|
| M1 | 68.47 | 1.91 ↓ |
| M2 | 68.88 | 1.50 ↓ |
| M3 | 69.30 | 1.08 ↓ |
| **Fed-MoLE** | **70.38** | - |

show significant performance degradation, as the fine-grained instance-level differences captured by the routed LoRA experts are diminished during global aggregation. Meanwhile, the results of M2 indicate that aligning cross-client knowledge benefits the enhancement of the global representation.

**Hyper-parameter Analysis.** To thoroughly investigate the impact of two key hyper-parameters, we tune $K_r$ and $N$ from $\{1, 2, 4, 8, 12\}$, subject to $K_r \leq N$. We report the average performance, and the results are presented in Figure 3. As observed, using more routed LoRA experts improves the performance of Fed-MoLE, as a larger number of experts can better capture instance-level differences at a finer granularity, thereby more effectively handling the diverse data distributions across clients. However, increasing $K_r$ does not always lead to performance improvement, as it may reduce the specialization benefits of individual experts. Fed-MoLE achieves the best performance when $K_r = 2$ with 8 experts.

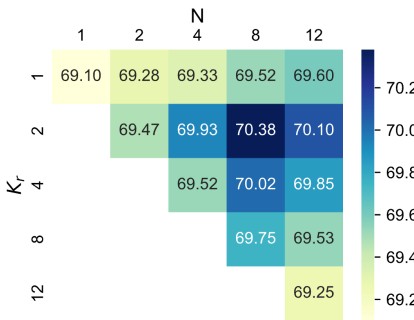

Figure 3: **Hyper-parameter analysis** of Fed-MoLE with different $K_r$ and $N$ on Fed-VQA.

### 4.4 Efficiency Analysis

**Communication Cost.** Table 5 presents the communication overhead. Apparently, fully fine-tuning of LLaVA-1.5-7B across 5 clients incurs a per-round communication cost of **141.5 GB**, rendering it impractical for real-world deployment. In contrast, LoRA reduces the cost to **226.8 MB** (only **0.16%**), significantly improving communication efficiency. Compared to FedAvg+LoRA, Fed-MoLE introduces no additional communication overhead, as it aligns only a single LoRA adapter across clients to obtain a unified global representation.

**Computation Cost.** We further present the peak memory usage during training in Table 5. The multiple parallel LoRA experts in Fed-MoLE exhibit sparsity, with only $K_r$ experts being activated. Even when using 8 additional routed LoRA experts, the peak memory during training increases by only 2.37 GB, allowing Fed-MoLE to be deployed under the same resource conditions as FedAvg+LoRA. Compared to the significant accuracy improvement, this memory overhead is acceptable.

Table 5: **Efficiency analysis** on Fed-VQA using LLaVA-1.5-7B as MLLMs, reporting per-round communication cost of five clients and peak memory. 'OOM' represents out-of-memory.

| Method | MLLMs | Cost | Memory |
|---|---|---|---|
| FedAvg + Full Fine-tune | 14.15 GB | 141.5 GB | OOM |
| FedAvg + LoRA | 14.15 GB | 226.8 MB | 14.38 GB |
| **Fed-MoLE** | 14.15 GB | 226.8 MB | 16.75 GB |

## 5 Conclusion

In this work, we identify a critical yet overlooked issue in federated fine-tuning of MLLMs under non-IID data: FL can fail to outperform standalone local training, *i.e.*, "$1 + 1 < 1$". To address this, we propose Fed-MoLE, which adopts a hybrid MoE architecture combined with an alternating disentanglement–alignment mechanism to handle diverse data distributions across clients. Extensive experiments on two benchmarks demonstrate that Fed-MoLE consistently outperforms Standalone in both local and global performance, effectively overcoming the $1 + 1 < 1$" barrier. Furthermore, Fed-MoLE surpasses a range of SOTA FL baselines, further highlighting the superiority of our approach. We believe that Fed-MoLE advances the practical deployment of federated fine-tuning of MLLMs in real-world scenarios.

ETHICS STATEMENT

All authors have read and adhered to the ICLR Code of Ethics. Our work does not involve human subjects, sensitive personal data, or potentially harmful applications. All datasets used in this study are publicly available and widely used in prior research. We have ensured fairness and non-discrimination in data usage and methodology, and there are no conflicts of interest or ethical concerns related to sponsorship.

REPRODUCIBILITY STATEMENT

We have taken steps to ensure reproducibility of our results. Details of the model architectures, training procedures, datasets, and evaluation metrics are provided in the main text (Sec. 4.1) and appendix (Sec. A). All datasets used are publicly available. The source code will be released upon acceptance to further facilitate reproducibility.

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

Table 6: **Number of samples** in the `train` and `test` sets for the two benchmarks.

| Split | Fed-VQA | | | | | Fed-Med | | |
|---|---|---|---|---|---|---|---|---|
| | AQUA | GQA | VizWiz | Abstract | COCO-QA | VQA | report generation | detection |
| `train` | 2518 | 2534 | 2493 | 2476 | 2487 | 2979 | 6558 | 6000 |
| `test` | 503 | 533 | 499 | 508 | 509 | 200 | 200 | 200 |

# A EXPERIMENTAL DETAILS

## A.1 DATASETS

We conducted experiments on two benchmarks, Fed-VQA (Chen et al., 2024) and Fed-Med (Zheng et al., 2025). In particular, Fed-VQA is composed of five clients, where each client corresponds to a different VQA dataset, as described below:

- **AQUA** (Garcia et al., 2020): It is designed for visual question answering on artworks and includes two types of questions: visual questions grounded in the paintings and knowledge-based questions derived from their accompanying commentary.

- **GQA** (Hudson & Manning, 2019): It provides large-scale visual question answering tasks grounded in real-world images, designed to evaluate models' abilities in compositional reasoning over objects, attributes, and relationships within complex scenes.

- **VizWiz** (Gurari et al., 2018): It focuses on visual question answering for images captured by blind people, presenting challenges such as image quality issues, object occlusion, and diverse real-world content.

- **Abstract** (Antol et al., 2015): It provides visual question answering tasks over cartoon-like or synthetic scenes, allowing models to focus on reasoning about object interactions and spatial relationships without the complexity of real-world image variability.

- **COCO-QA** (Ren et al., 2015): It is derived from MS COCO images and provides visual question answering tasks, where questions focus on objects, numbers, colors, and locations within everyday scenes.

Fed-Med consists of three medical clients, each focusing on a different task. The clients correspond to VQA, report generation, and detection, with their details provided below:

- **VQA**: The data of this client is from two medical VQA datasets, including SLAKE (Liu et al., 2021) and VQA-RAD (Lau et al., 2018). SLAKE focuses on diverse clinical scenarios with a wide range of question types, while VQA-RAD contains radiology images paired with questions that require understanding medical content and terminology.

- **Report generation:** The data of this client is from the MIMIC-CXR (Johnson et al., 2019) dataset, which is a large-scale chest X-ray dataset paired with textual radiology reports, supporting tasks such as report generation, abnormality detection, and clinical reasoning.

- **Detection:** The data of this client is from the RadGenome-Chest CT (Zhang et al., 2024a) dataset, which is a large-scale chest CT dataset annotated for multiple abnormalities, supporting tasks such as detection, segmentation, and clinical reasoning. We need to obtain the bounding box coordinates of the lesions given the CT images.

For both Fed-VQA and Fed-Med, each client's dataset is divided into `train` and `test` sets, with the number of samples listed in Table 6. The `train` sets are used for federated fine-tuning of the MLLMs, and the fine-tuned models are evaluated on each client's `test` set to report the testing results. In Fed-VQA, clients' data differ in both text and images, including variations in question types such as yes/no, counting, or open-ended questions, as well as differences in image styles, such as cartoon, artistic, and real-world images. In contrast, Fed-Med not only exhibits distribution differences in image and text across clients but also introduces task-level heterogeneity.

## A.2 EVALUATION

The two benchmarks involve three different tasks: VQA, report generation, and detection. For each task, we adopt task-specific evaluation metrics. VQA aims to generate the correct answer given an image and a question. Since the ground-truth answers in the VQA tasks of the benchmarks are all single-word responses, we use accuracy (%) as the evaluation metric, counting a prediction as correct only when it exactly matches the ground-truth answer. Detection focuses on predicting the bounding box coordinates of lesions in medical images, and performance is evaluated using the Intersection over Union (IoU, %) between ground-truth and predicted boxes. Report generation aims to produce diagnostic reports from patients' medical images. Due to the complexity of report content, we employ a widely used LLM-based judging approach, using Qwen-3-8B as the judge model to assess the quality of generated reports against the ground-truth. The input template provided to the judge model is as follows:

---

You are a medical expert evaluating the quality of an automatically generated radiology report based on a reference ground-truth report. Your goal is to provide a score from 1 to 5 (integer only) and a concise justification based on the following criteria.

1. Clinical Accuracy: Does the generated report correctly identify and describe the key findings and pathologies that appear in the ground-truth report? Are any major findings missing or hallucinated?
2. Completeness: Does the generated report cover all the important aspects included in the ground-truth report?
3. Fluency and Readability: Is the generated report grammatically correct, well-structured, and easy to read?

Please follow these steps:
- Read both the Generated Report and the Reference Report.
- Provide a score from 1 (poor) to 5 (excellent) reflecting the overall quality of the generated report in comparison to the reference, mainly focusing on clinical accuracy.
- Provide a short explanation (2–4 sentences) justifying your score.

Format your output strictly as follows:
Output format:
Score: `<Score>`
Reason: `<brief explanation>`

Here is the input for this evaluation:
Generated Report: `<prediction>`
Reference Report: `<ground-truth>`

---

where `<prediction>` and `<ground-truth>` correspond to the model-generated report and the ground-truth report, respectively. The judge outputs a final `<score>` (%) on a scale from 100% (poor) to 500% (excellent), along with a brief explanation of the judgment.

## B ADDITIONAL EXPERIMENTS

### B.1 CONVERGENCE

Figure 4 shows the training loss of `Standalone`, FedAvg and Fed-MoLE over communication rounds on five clients. It can be observed that under non-IID data, `Standalone` achieves better convergence on each client compared to `FedAvg`, as client drift can significantly affect convergence. Since Fed-MoLE can effectively handle diverse data distributions across clients, it achieves convergence performance comparable to `Standalone`, further demonstrating the effectiveness of our method for federated fine-tuning of MLLMs under non-IID data.

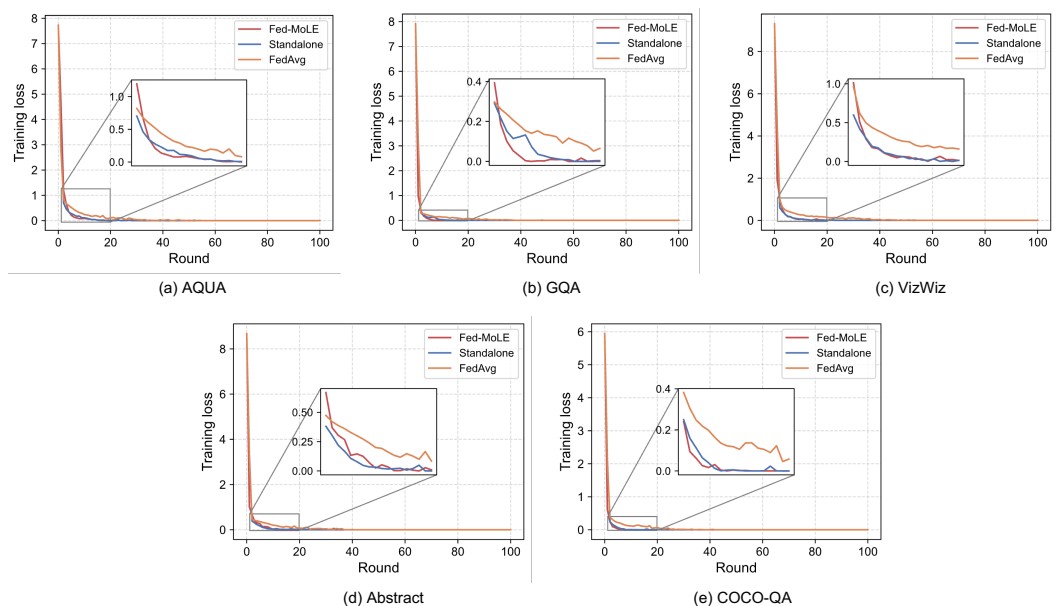

Figure 4: **Illustration of training loss** versus communication rounds on Fed-VQA.

Table 7: **Quantitative comparison of all methods** on **Fed-VQA** (Chen et al., 2024) benchmark using **Qwen2-VL-7B** (Wang et al., 2024a) as foundation model comprising five clients: AQUA (Garcia et al., 2020), GQA (Hudson & Manning, 2019), VizWiz (Gurari et al., 2018), Abstract (Antol et al., 2015), and COCO-QA (Ren et al., 2015). We report the test accuracy (%) of each method on every client's dataset, along with the average (Avg.) across all clients. The best results are marked in **bold**.

| Method | AQUA Acc(%) | Δ | GQA Acc(%) | Δ | VizWiz Acc(%) | Δ | Abstract Acc(%) | Δ | COCO-QA Acc(%) | Δ | Avg.(%) | Δ |
|---|---|---|---|---|---|---|---|---|---|---|---|---|
| Standalone | 68.78 | - | 76.17 | - | 62.72 | - | **80.51** | - | 83.30 | - | 74.29 | - |
| *Global Model Learning* | | | | | | | | | | | | |
| FedAvg | 70.57 | 1.79 ↑ | 74.10 | 2.07 ↓ | 62.32 | 0.40 ↓ | 75.00 | 5.51 ↓ | 82.12 | 1.18 ↓ | 72.82 | 1.47 ↓ |
| FedProx | **72.76** | 3.98 ↑ | 73.17 | 3.00 ↓ | 62.32 | 0.40 ↓ | 70.86 | 9.65 ↓ | 76.42 | 6.88 ↓ | 71.10 | 3.19 ↓ |
| FedAdam | **72.76** | 3.98 ↑ | 74.48 | 1.69 ↓ | 61.92 | 0.80 ↓ | 74.06 | 6.45 ↓ | 82.51 | 0.79 ↓ | 73.25 | 1.04 ↓ |
| FedYogi | 71.96 | 3.18 ↑ | 72.98 | 3.19 ↓ | 59.31 | 3.41 ↓ | 76.18 | 4.03 ↓ | 83.10 | 0.20 ↓ | 72.71 | 1.58 ↓ |
| FedAdagrad | 71.76 | 2.98 ↑ | 75.42 | 0.75 ↓ | 59.91 | 2.81 ↓ | 75.00 | 5.51 ↓ | 84.08 | 0.78 ↑ | 73.23 | 1.06 ↓ |
| Scaffold | 71.56 | 2.78 ↑ | 75.23 | 0.94 ↓ | 63.52 | 0.80 ↑ | 74.80 | 5.71 ↓ | 83.58 | 0.28 ↑ | 73.73 | 0.56 ↓ |
| FFA-LoRA | 65.00 | 3.78 ↓ | 69.98 | 6.19 ↓ | 55.11 | 7.61 ↓ | 72.83 | 7.68 ↓ | 77.40 | 5.90 ↓ | 68.06 | 6.23 ↓ |
| FLexLoRA | 70.77 | 1.99 ↑ | 74.67 | 1.50 ↓ | 60.72 | 2.00 ↓ | 76.37 | 4.14 ↓ | 82.31 | 0.99 ↓ | 72.97 | 1.32 ↓ |
| FRLoRA | 71.98 | 3.20 ↑ | 74.67 | 1.50 ↓ | 63.72 | 1.00 ↑ | 74.64 | 5.87 ↓ | 83.12 | 0.18 ↓ | 73.62 | 0.67 ↓ |
| *Personalized Model Learning* | | | | | | | | | | | | |
| FedDPA-LoRA | 67.39 | 1.39 ↓ | 75.42 | 0.75 ↓ | 62.52 | 0.20 ↓ | 77.36 | 3.15 ↓ | 84.28 | 0.98 ↑ | 73.39 | 0.90 ↓ |
| FedSA-LoRA | 71.96 | 3.18 ↑ | 74.85 | 1.34 ↓ | **68.13** | 5.41 ↑ | 75.39 | 5.12 ↓ | 82.31 | 0.99 ↓ | 74.53 | 0.24 ↑ |
| **Fed-MoLE** | 72.36 | 3.58 ↑ | **76.73** | 0.56 ↑ | 65.13 | 2.41 ↑ | 77.16 | 3.35 ↓ | **85.46** | 2.16 ↑ | **75.36** | 1.07 ↑ |

## B.2 DIFFERENT MLLMS

To further evaluate the robustness and effectiveness of Fed-MoLE, we conducted additional experiments on Fed-VQA using a different MLLM, Qwen2-VL-7B (Wang et al., 2024a), as the base model. As shown in Table 7, all methods achieved performance improvements with this stronger MLLM. More importantly, compared to Standalone, nearly all clients in Fed-MoLE benefited from collaborative training, resulting in gains in both local and global performance. Furthermore, Fed-MoLE consistently outperformed all other FL methods. This consistent superior performance demonstrates the robustness and effectiveness of Fed-MoLE across different MLLMs.

Figure 5: **Qualitative Comparison** of Fed-MoLE *vs.* Standalone on Fed-VQA.

## B.3 QUALITATIVE COMPARISON

Figure 5 shows the results of a qualitative comparison between Fed-MoLE and Standalone on all clients of Fed-VQA. Apparently, different clients exhibit distribution differences in both image and text. Fed-MoLE can disentangle diverse instance-level variations and align cross-client knowledge into a unified global representation, yielding superior performance over Standalone.

## B.4 PERFORMANCE UNDER IID DATA

In this section, we briefly explore the performance of Fed-MoLE under IID settings. To construct the IID scenario, we randomly partition the training and test sets of the AQUA (Garcia et al., 2020) dataset into five clients, ensuring that each client contains the same number of samples. We evaluate the trained model on the test set of each client and report the average accuracy in Table 8. It can be observed that under the IID setting, the "$1 + 1 < 1$" issue does not occur. Moreover, by leveraging data from multiple clients, FedAvg achieves better performance than Standalone. In addition, Fed-MoLE still significantly outperforms both FedAvg and Standalone under IID settings, as it can effectively handle instance-level differences.

Table 8: **Results** of Standalone, FedAvg and Fed-MoLE on AQUA (Garcia et al., 2020) under IID setting.

| Method | Avg. |
|---|---|
| Standalone | 70.57 |
| FedAvg | 71.76 |
| **Fed-MoLE** | 73.36 |

## B.5 ANALYSIS OF ALTERNATING DISENTANGLEMENT−ALIGNMENT

In this section, we analyze the cross-client feature differences learned by the shared LoRA and routed LoRA experts. Before aggregation, we compute the pairwise cosine similarity between clients using the outputs of the two paths in the final tuned layer of Fed-MoLE, as well as the corresponding layer in FedAvg. The averaged results on Fed-VQA are presentd in Table 9.

Compared to FedAvg, the shared LoRA path exhibits substantially higher cross-client similarity, whereas the routed LoRA path shows much lower similarity. This indicates that the shared LoRA expert captures knowledge that are common across samples and clients, while the routed LoRA experts models instance-level, high-variance local variations. These results demonstrate that Fed-MoLE effectively disentangles global and local information, so that aggregating only the shared LoRA can reduce client drift while effectively integrating global knowledge to improve model generalization.

Table 9: **Analysis of cross-client feature differences** in the Shared vs. Routed LoRA paths of Fed-MoLE on FedVQA (Chen et al., 2024).

| Fed-MoLE | | FedAvg |
|---|---|---|
| Shared LoRA path | Routed LoRA path | **FedAvg** |
| 0.823 | 0.354 | 0.679 |

## B.6 PERFORMANCE UNDER DIFFERENT LoRA RANKS

Table 10: **Quantitative comparison** on **Fed-VQA** (Chen et al., 2024) benchmark using LLaVA-1.5-7B as foundation model with different rank $r$ of LoRA. We report the test accuracy (%) of each method on every client's dataset, along with the average (Avg.) across all clients. The best results are marked in **bold**.

| | AQUA | | GQA | | VizWiz | | Abstract | | COCO-QA | | | |
|---|---|---|---|---|---|---|---|---|---|---|---|---|
| **Method** | Acc(%) | $\Delta$ | Acc(%) | $\Delta$ | Acc(%) | $\Delta$ | Acc(%) | $\Delta$ | Acc(%) | $\Delta$ | Avg.(%) | $\Delta$ |
| | | | | | | $r = 16$ | | | | | | |
| Standalone | 71.66 | - | 72.10 | - | 62.13 | - | 67.11 | - | 80.94 | - | 70.78 | - |
| FedAvg | 70.37 | 1.29 ↓ | 70.11 | 1.99 ↓ | 61.12 | 1.01 ↓ | 66.92 | 0.19 ↓ | 77.60 | 3.34 ↓ | 69.22 | 1.56 ↓ |
| FedSA-LoRA | 70.95 | 0.71 ↓ | **72.60** | 0.50 ↑ | 62.32 | 0.19 ↑ | 67.91 | 0.80 ↑ | 80.72 | 0.22 ↓ | 70.91 | 0.13 ↑ |
| **Fed-MoLE** | **74.37** | 2.71 ↑ | 72.42 | 0.32 ↑ | **63.92** | 1.79 ↑ | **68.27** | 1.16 ↑ | **81.53** | 0.59 ↑ | **72.10** | 1.32 ↑ |
| | | | | | | $r = 32$ | | | | | | |
| Standalone | 72.36 | - | 73.73 | - | 62.92 | - | 65.94 | - | 80.06 | - | 71.00 | - |
| FedAvg | 71.76 | 0.60 ↓ | 71.66 | 2.07 ↓ | 58.71 | 4.21 ↓ | 68.70 | 2.76 ↑ | 79.37 | 0.69 ↓ | 70.04 | 0.96 ↓ |
| FedSA-LoRA | 72.55 | 0.19 ↑ | 72.98 | 0.75 ↑ | 62.32 | 0.60 ↑ | 68.29 | 2.35 ↑ | 81.53 | 1.47 ↓ | 71.54 | 0.54 ↑ |
| **Fed-MoLE** | **74.95** | 2.59 ↑ | **74.29** | 0.56 ↑ | **63.12** | 0.20 ↑ | **69.09** | 3.15 ↑ | **81.72** | 1.66 ↑ | **72.63** | 1.63 ↑ |

To further investigate the impact of $r$, we conduct additional experiments on Fed-VQA with $r = 16$ and $r = 32$, where $\alpha$ is set to 32 and 64 respectively to keep $\frac{\alpha}{r} = 2$. The results are reported in Table 10. As we can see, the $1 + 1 < 1$ phenomenon consistently appears across all settings, confirming that it does not arise from the LoRA configuration. Moreover, across all LoRA setups, Fed-MoLE consistently outperforms standalone on every client and surpasses FedSA-LoRA in terms of global performance, demonstrating the robustness and generality of our method.

## C   THE USE OF LARGE LANGUAGE MODELS

We only used large language models (LLMs) for minor tasks such as grammar checking and expression polishing. The models did not contribute to research ideation, experimental design, analysis, or drafting of technical content. All scientific claims, experimental results, and writing remain the sole responsibility of the authors.

## D   GENERALIZATION BOUND

In this section, we analyze the generalization bound of our method. For clarity, we denote the MoLE structure by a function $h\colon x \to y$, where $h \in \mathcal{H}$, $d$ represents the VC-dimension of the hypothesis space $\mathcal{H}$. In addition, we make the following assumption regarding the loss function.

**Assumption 1.** *(Bounded Cross-Entropy Loss) We assume the predicted class probabilities satisfy*

$$p(x; \omega) \geq \theta, \quad \forall x \tag{12}$$

*for small constant $\theta > 0$.*

The above assumption ensures that the cross-entropy loss is bounded:

$$0 \leq \ell \leq log\frac{1}{\theta}. \tag{13}$$

Besides, we introduce McDiarmid's Inequality as theoretical support.

**Lemma 1.** *(McDiarmid's Inequality) Let $S = (X_1, \ldots, X_m) \in \mathcal{X}^m$ be a set of independent random variables. Assume that there exist constants $c_1, \ldots, c_m > 0$ and a function $f : \mathcal{X}^m \to \mathbb{R}$ such that for all $x_1, \ldots, x_m, x_i' \in \mathcal{X}^m$ and for every $i \in \{1, \ldots, m\}$,*

$$\sup_{x_i' \in \mathcal{X}^m} \left| f(x_1, \ldots, x_i, \ldots, x_m) - f(x_1, \ldots, x_i', \ldots, x_m) \right| \leq c_i. \tag{14}$$

*Then, for any $\epsilon > 0$,*

$$\mathbb{P}(f(S) - \mathbb{E}[f(S)] \geq \epsilon) \leq \exp\left( -\frac{2\epsilon^2}{\sum_{i=1}^n c_i^2} \right). \tag{15}$$

Using this lemma, we can establish the generalization bound for our method as follows:

**Theorem 1.** *(Generalization Bound) Consider $K$ clients with their empirical data distributions $D_1, D_2, \ldots, D_K$, and $n_k$ is the number of samples in $D_k$, and $\mathcal{N} = \sum_{i=1}^{K} n_i$. Let $\hat{D}_1, \hat{D}_2, \ldots, \hat{D}_K$ denote the corresponding true data distributions. Then, with probability at least $1 - \delta$, we have*

$$\sup \left| \sum_{k=1}^{K} \frac{n_k}{\mathcal{N}} \mathcal{L}_{D_k}(\cdot) - \sum_{k=1}^{K} \frac{n_k}{\mathcal{N}} \mathcal{L}_{\hat{D}_k}(\cdot) \right| \leq \left( \log \frac{1}{\theta} \right) \sqrt{\frac{1}{2\mathcal{N}} \log \frac{1}{\delta}} + \sqrt{\frac{d}{\mathcal{N}} \log \frac{e\mathcal{N}}{d}}. \quad (16)$$

*Proof.* Equation (15) equals to

$$\mathbb{P}(f(S) - \mathbb{E}[f(S)] \leq \epsilon) \leq 1 - \exp\left( -\frac{2\epsilon^2}{\sum_{i=1}^{m} c_i^2} \right). \quad (17)$$

This means that with probability at least $1 - \exp\left( -\frac{2\epsilon^2}{\sum_{i=1}^{m} c_i^2} \right)$,

$$f(S) - \mathbb{E}[f(S)] \leq \epsilon. \quad (18)$$

Let $\delta = \exp\left( -\frac{2\epsilon^2}{\sum_{i=1}^{n} c_i^2} \right)$, the above can be rewritten as with probability at least $1 - \delta$,

$$f(S) - \mathbb{E}[f(S)] \leq \sqrt{\frac{\sum_{i=1}^{m} c_i^2}{2} \log \frac{1}{\delta}}. \quad (19)$$

With Assumption 1, the loss is bounded within $[0, \log \frac{1}{\theta}]$. Replacing a single sample from client $k$ affects the empirical mean loss at that client by at most $\frac{\log \frac{1}{\theta}}{n_k}$, which is scaled by the aggregation weight $\frac{n_k}{\mathcal{N}}$. Therefore, the impact of replacing one sample is bounded by

$$\sup_{x_i' \in \mathcal{X}^{\mathcal{N}}} \left| \left( \sum_{k=1}^{K} \frac{n_k}{\mathcal{N}} \mathcal{L}_{D_k}(x_i; \cdot) - \sum_{k=1}^{K} \frac{n_k}{\mathcal{N}} \mathcal{L}_{\hat{D}_k}(x_i; \cdot) \right) - \left( \sum_{k=1}^{K} \frac{n_k}{\mathcal{N}} \mathcal{L}_{D_k}(x_i'; \cdot) - \sum_{k=1}^{K} \frac{n_k}{\mathcal{N}} \mathcal{L}_{\hat{D}_k}(x_i'; \cdot) \right) \right|$$

$$\leq c_i \leq \frac{\log \frac{1}{\theta}}{n_k} \cdot \frac{n_k}{\mathcal{N}} \leq \frac{\log \frac{1}{\theta}}{\mathcal{N}}. \quad (20)$$

Consequently, by substituting $f(\cdot)$ with $\left( \sum_{k=1}^{K} \frac{n_k}{\mathcal{N}} \mathcal{L}_{D_k}(\cdot) - \sum_{k=1}^{K} \frac{n_k}{\mathcal{N}} \mathcal{L}_{\hat{D}_k}(\cdot) \right)$, we obtain that, with probability at least $1 - \delta$,

$$\left( \sum_{k=1}^{K} \frac{n_k}{\mathcal{N}} \mathcal{L}_{D_k}(\cdot) - \sum_{k=1}^{K} \frac{n_k}{\mathcal{N}} \mathcal{L}_{\hat{D}_k}(\cdot) \right) - \mathbb{E} \left[ \left( \sum_{k=1}^{K} \frac{n_k}{\mathcal{N}} \mathcal{L}_{D_k}(\cdot) - \sum_{k=1}^{K} \frac{n_k}{\mathcal{N}} \mathcal{L}_{\hat{D}_k}(\cdot) \right) \right]$$

$$\leq \sqrt{\frac{\sum_{i=1}^{\mathcal{N}} c_i^2}{2} \log \frac{1}{\delta}}$$

$$\leq \sqrt{\frac{\sum_{i=1}^{\mathcal{N}} (\frac{\log \frac{1}{\theta}}{\mathcal{N}})^2}{2} \log \frac{1}{\delta}} \quad (21)$$

$$\leq \left( \log \frac{1}{\theta} \right) \sqrt{\frac{1}{2\mathcal{N}} \log \frac{1}{\delta}}.$$

Based on the Rademacher complexity and Jensen's inequality, we have

$$\mathbb{E}\left[\left(\sum_{k=1}^{K}\frac{n_k}{\mathcal{N}}\mathcal{L}_{D_k}(\cdot) - \sum_{k=1}^{K}\frac{n_k}{\mathcal{N}}\mathcal{L}_{\hat{D}_k}(\cdot)\right)\right]$$

$$\leq \sum_{k=1}^{K}\frac{n_k}{\mathcal{N}}\left(\mathcal{L}_{D_k}(\cdot) - \mathcal{L}_{\hat{D}_k}(\cdot)\right)$$

$$\leq \underbrace{\sum_{k=1}^{K}\frac{n_k}{\mathcal{N}}\mathfrak{R}_k(\mathcal{H}) = \sum_{k=1}^{K}\frac{n_k}{\mathcal{N}}\mathbb{E}\left[\sup_{h\in\mathcal{H}}\frac{1}{n_k}\sum_{i=1}^{n_k}\sigma_i h(x_i)\right]}_{\text{Rademacher complexity}} \tag{22}$$

$$\leq \sum_{k=1}^{K}\frac{n_k}{\mathcal{N}}\sqrt{\frac{d}{n_k}\log\frac{en_k}{d}}$$

$$\leq \sum_{k=1}^{K}\frac{n_k}{\mathcal{N}}\sqrt{\frac{d}{n_k}\log\frac{e\mathcal{N}}{d}}$$

$$\leq \underbrace{\sqrt{\frac{d}{\mathcal{N}}\log\frac{e\mathcal{N}}{d}}}_{\text{Jensen's inequality}},$$

where $\sigma_i$ is independent random variable uniformly distributed over $\{-1, 1\}$.

Finally, by combing Equations (21) and (22), we have

$$\sup\left|\sum_{k=1}^{K}\frac{n_k}{\mathcal{N}}\mathcal{L}_{D_k}(\cdot) - \sum_{k=1}^{K}\frac{n_k}{\mathcal{N}}\mathcal{L}_{\hat{D}_k}(\cdot)\right| \leq \left(\log\frac{1}{\theta}\right)\sqrt{\frac{1}{2\mathcal{N}}\log\frac{1}{\delta}} + \sqrt{\frac{d}{\mathcal{N}}\log\frac{e\mathcal{N}}{d}}. \tag{23}$$

