# OpenReview forum: "$1+1<1$? Breaking the Standalone Barrier in Federated Fine-Tuning of Multimodal Large Language Models under Non-IID Data"
_ICLR.cc/2026/Conference — Submitted to ICLR 2026_

### Official Review · Reviewer_2zi3 · 2025-10-30

**Soundness:** 2
**Presentation:** 2
**Contribution:** 2
**Rating:** 4
**Confidence:** 5

**Summary:**

This paper focuses on a fundamental issue in federated fine-tuning: under non-IID data, some federated fine-tuning methods perform even worse than standalone local training. To enhance the effectiveness of federated learning, the authors introduce a Mixture of Experts (MoE) framework. Specifically, each client includes both shared and specialized experts, which are trained alternately, the specialized experts capture instance-level variations, while the shared expert integrates common knowledge across clients. Experiments on multiple datasets demonstrate that the proposed method not only significantly improves federated fine-tuning performance but also successfully overcomes the limitation where federated training underperforms local training.

**Strengths:**

This method reveals that existing federated fine-tuning approaches can sometimes perform worse than standalone local training, a finding that is highly significant for federated learning.

**Weaknesses:**

1. The method shows limited innovation, as applying MoE to federated LoRA fine-tuning has already been explored in several studies (e.g., [a–d]); the main difference here seems to lie only in the use of an alternating training strategy.

[a] Le, Khiem, et al. "FLAME: Towards Federated Fine-Tuning Large Language Models Through Adaptive SMoE." arXiv preprint arXiv:2506.16600 (2025).
[b] Wang, Lei, et al. "Adaptive LoRA Experts Allocation and Selection for Federated Fine-Tuning." arXiv preprint arXiv:2509.15087 (2025).
[c] Hu, Gang, et al. "FFT-MoE: Efficient Federated Fine-Tuning for Foundation Models via Large-scale Sparse MoE under Heterogeneous Edge." arXiv preprint arXiv:2508.18663 (2025).
[d] Chen, Fahao, et al. "Federated Fine-Tuning of Sparsely-Activated Large Language Models on Resource-Constrained Devices." arXiv preprint arXiv:2508.19078 (2025).

2. As shown in Table 1, the performance improvement over FedSA-LoRA is relatively small. The introduction of MoE in clients provides only marginal gains while increasing computational and communication costs.
3. Although the discovery of the “1 + 1 < 1” phenomenon is interesting, the paper lacks further investigation into its causes — for example, whether data heterogeneity or LoRA configuration contributes to the inferior performance of federated LoRA fine-tuning compared to local training.
4. Additional ablation studies are recommended for the client-side MoE design in LoRA, such as analyzing the impact of the number of specialized experts and the rank of LoRA on model performance.

**Questions:**

The main issue lies in the lack of methodological innovation. Many studies have already applied MoE to federated LoRA fine-tuning, and this work merely adds an alternating training strategy. The performance improvement is limited, and the ablation studies are insufficient.

---

> ### Author Response · Authors · 2025-11-21
> **Official Comment by Authors**
>
> > W1. The method shows limited innovation.
>
> We respectfully clarify that [a–d] appeared **contemporaneously** with ours, and their goals and designs **differ substantially** from Fed-MoLE.
>
> These studies simply introduce **vanilla** MoE-LoRA (i.e., all LoRA experts routed by a router) into the FL, which naturally leads to heterogeneous activation patterns across clients. To address the resulting **incompatibility between MoE-LoRA and FL**, they adopt auxiliary mechanisms such as activation-frequency aggregation [a], clustering-based aggregation [b], or additional loss terms [c]. [d] focuses on optimizing MoE-based LLMs in FL, which is **fundamentally different** from our topic.
>
> In contrast, the novelty of Fed-MoLE does **not** lie in the use of MoE. Our key contribution is **the first** to reveal the “1+1 < 1” phenomenon in federated fine-tuning of MLLMs, where naive FedLoRA can perform even worse than single-client training. Motivated by this, we introduce a **hybrid MoE+LoRA** architecture consisting of a shared LoRA expert that integrates cross-client knowledge and dynamically routed LoRA experts that capture instance-level, high-variance local patterns. Therefore, Fed-MoLE introduces **a new perspective and architectural formulation** rather than a straightforward application of MoE to federated LoRA fine-tuning.
>
>
> > W2. Slight improvement in Table 1.
>
> (1) Prior FL studies [4-5] **typically** use global performance (Avg.) as the primary evaluation metric, as this **better** reflects the model's generalization capability. In Table 1, Fed-MoLE outperforms FedSA-LoRA in terms of global performance. Besides, we further conducted three independent trials using different random seeds on Fed-VQA, and reported the mean and standard deviation. Furthermore, we assessed the significance of the performance differences using the paired t-test. The results are shown below:
>
> | Method | Avg. | p-value |
> | :-----|:----: |:----: |
> | FedSA-LoRA | $69.62 \pm 0.52$ |  - |
> | Fed-MoLE  | $70.84 \pm 0.36$ | 0.011 |
>
> The experimental results indicate that Fed-MoLE shows a **significant improvement** over FedSA-LoRA in global performance (p-value < 0.05)
>
> (2) FedSA-LoRA indeed shows a strong advantage on individual clients, as its personalization strategy allows it to better fit each client’s data. However, it does **not** balance all clients well, as evidenced by its **lower** performance than the standalone model on AQUA and GQA. In contrast, Fed-MoLE is the **only** method that outperforms the standalone model across all clients, indicating that it enables all clients to benefit from FL, which aligns **better** with practical value.
>
>
> > W3. Causes of “1 + 1 < 1” phenomenon.
>
> We confirm that the “1+1 < 1” phenomenon is **not** caused by the LoRA configuration, as both Standalone and FedLoRA use **exactly the same LoRA configuration**. As shown in Figure 1 (b) and Figure 1 (c), the phenomenon is from client drift under non-IID data, which leads to a representation mismatch between the aggregated global model and local models. Moreover, our IID experiments in **Table 8** demonstrate that this phenomenon does not occur when data distributions are homogeneous, further verifying that it is induced by data heterogeneity rather than by the LoRA configuration.

---

> ### Author Response · Authors · 2025-11-21
> **Official Comment by Authors**
>
> > W4. Additional ablation studies.
>
> (1) The ablation on the number of specialized experts is presented in **Figure 3**. The results show that increasing the number of experts brings **only** marginal improvements, indicating that the gains do **not** come from scaling up the MoE. **Instead**, they arise from Fed-MoLE’s architecture and its alternating decoupled-alignment mechanism, which effectively alleviates data heterogeneity. This further suggests that Fed-MoLE’s core contribution lies **not** in stacking more experts, but in **providing a new perspective** for balancing global generalization and local adaptation.
>
> (2) We further conduct an ablation study with different values of $r$, and the results are shown below. The 1+1<1 phenomenon consistently appears across all settings, confirming that it does not arise from the LoRA configuration. Moreover, across all LoRA setups, Fed-MoLE consistently outperforms standalone on every client and surpasses FedSA-LoRA in terms of global performance, demonstrating the **robustness and generality** of our method.
>
>
> *Table 1: r=8*
> | Method      | AQUA   |GQA  | VizWiz  | Abstract   | COCO-QA | Avg. |
> |-------------|-------|-------|-------|--------|--------|--------|
> | Standalone    | 71.17 | 71.54 | 60.92| 63.77| 78.78| 69.04|
> |FedAvg| 68.59 |70.29 |62.32 |61.22 | 76.42 |67.77|
> |FedSA-LoRA| 70.97| 69.98| 63.72| 64.56| 79.96|69.84|
> | **Fed-MoLE (Ours)** | 74.95 | 71.10 | 61.32 | 64.96 | 79.37 | **70.38** |
>
>
> *Table 2: r=16*
> | Method      | AQUA   |GQA  | VizWiz  | Abstract   | COCO-QA | Avg. |
> |-------------|-------|-------|-------|--------|--------|--------|
> | Standalone    | 71.66 | 72.10 | 62.13| 67.11| 80.94| 70.78|
> |FedAvg| 70.37 |70.11 |61.12 |66.92 | 77.60 |69.22|
> |FedSA-LoRA| 70.95 | 72.60 |62.32 |67.91 |80.72 |70.91|
> | **Fed-MoLE (Ours)** | 74.37 |72.42 |63.92 |68.27 |81.53 |72.10|
>
> *Table 3: r=32*
> | Method      | AQUA   |GQA  | VizWiz  | Abstract   | COCO-QA | Avg. |
> |-------------|-------|-------|-------|--------|--------|--------|
> | Standalone    | 72.36|73.73 |62.92 |65.94 | 80.06| 71.00|
> |FedAvg| 71.76|71.66|58.71 |68.70 |79.37 |70.04|
> |FedSA-LoRA | 72.55 |72.98|62.32|68.29|81.53|71.54|
> | **Fed-MoLE (Ours)** |74.95|74.29|63.12|69.09|81.72|72.63|
>
>
> > Q1. The main issue lies in the lack of methodological innovation. Many studies have already applied MoE to federated LoRA fine-tuning, and this work merely adds an alternating training strategy. The performance improvement is limited, and the ablation studies are insufficient.
>
> The novelty of Fed-MoLE does **not** lie in the use of MoE. Our key contribution is **the first** to reveal the “1+1 < 1” phenomenon in federated fine-tuning of MLLMs, where naive FedLoRA can perform even worse than single-client training. Motivated by this, we introduce a **hybrid MoE+LoRA** architecture consisting of a shared LoRA expert that integrates cross-client knowledge and dynamically routed LoRA experts that capture instance-level, high-variance local patterns. Therefore, Fed-MoLE introduces **a new perspective and architectural formulation** rather than a straightforward application of MoE to federated LoRA fine-tuning. We additionally conducted significance tests, confirming that the improvement is indeed statistically significant. The required ablation studies have been presented in the revised manuscript.

---

### Official Review · Reviewer_n4EL · 2025-10-31

**Soundness:** 3
**Presentation:** 3
**Contribution:** 2
**Rating:** 4
**Confidence:** 4

**Summary:**

In federated fine-tuning of MLLMs , the substantial communication overhead can be effectively mitigated through LoRA. Existing approaches typically allow all clients to share and collaboratively train a unified LoRA adapter. However, this paper finds that under non-IID conditions, federated fine-tuning often performs worse than training locally on each client, a phenomenon the authors refer to as “1+1 < 1.” To address this issue, the authors propose Fed-MoLE, a framework that employs dynamically routed LoRA experts to capture instance-level variations while utilizing a shared LoRA across all clients to extract a unified global representation. The two types of LoRA are trained in an alternating manner to achieve a balance between shared and personalized learning. Experiments on Fed-VQA and FedMed datasets demonstrate that Fed-MoLE achieves superior performance compared with existing methods.

**Strengths:**

1. Given the growing trend of data decentralization and the rapid development of large models, federated fine-tuning for MLLMs is a research topic of reasonable relevance.
2. The paper is generally well-structured, starting from an observed phenomenon and gradually introducing the proposed method in a logically coherent way.
3. The experiments are relatively complete, including multiple baseline comparisons that provide some validation for the proposed model’s effectiveness.

**Weaknesses:**

See Questions section.

**Questions:**

**Novelty**

- The logical connection between the reported “1+1 < 1” phenomenon and the proposed method is not sufficiently strong. The proposed framework still largely relies on the concept of shared parameters; overall, it appears more like introducing a direct comparison with *Standalone* in the experiments to highlight performance gains. It would be helpful for the authors to further clarify the causal relationship between this phenomenon and the design of their method.
- Although all local samples pass through the shared LoRA module, the paper does not provide a clear mechanism or experiment verifying that this module indeed captures cross-sample shared features. Likewise, there is no direct evidence showing how the routed LoRA models instance-level variations. The authors are encouraged to demonstrate the differences between the features captured by the two types of LoRA modules.
- Experimental results show that GFL typically underperforms *Standalone*, while PFL methods generally outperform *Standalone*. This suggests that under non-IID conditions, PFL methods may serve as a more appropriate comparison baseline. In this case, the emphasis on the “1+1 < 1” phenomenon may be less critical for evaluating the proposed method’s contribution.

**Cost**

- In line 268, the authors state, “This sparse structure significantly reduces the additional computational overhead when *N* is larger.” It is recommended to include quantitative results showing how computational cost varies with the number of experts $N$ and $K_r$.

**Details**

- The paper states that Fed-MoLE is applied only to the up_proj layer, while other layers use standard LoRA implementations. It would be helpful if the authors could explain the reasoning behind this design choice.
- Equation (6) may contain a typographical error; it should likely be written as $\sum^N_{i=1}\rho_iB^{r,t}_{k,i}A^{r,t}_{k,i}X$

---

> ### Author Response · Authors · 2025-11-21
> **Official Comment by Authors**
>
> > Q1. Relationship between “1+1 < 1” phenomenon and method.
>
> The root of the "1+1 < 1" phenomenon is the **significant divergence** in local LoRAs (Figure 1 (b)) across clients under non-IID data. Direct aggregation averages these conflicting local optima, which can lead to the representation mismatch between global and local LoRAs (Figure 1 (c)). This can degrade the global LoRA’s performance, sometimes even below that of single-client training. Fed-MoLE is explicitly designed to address this issue: it decouples the "**low-variance representation shared across clients**" (shared LoRA expert) from the "**high-variance, client- and sample-specific representation**" (routed LoRA experts), and aggregates only the shared representation. This **reduces** conflicts and client drift, enabling **stable** improvements under non-IID data. This establishes a clear **causal connection** between the "1+1 < 1" phenomenon and our method.
>
> > Q2. Disentanglement–alignment mechanism.
>
> We have analyzed the cross-client feature differences learned by the shared LoRA and routed LoRA experts. Before aggregation, we compute the pairwise cosine similarity between clients using the outputs of the two paths in the final tuned layer of Fed-MoLE, as well as the corresponding layer in FedAvg. The averaged results on Fed-VQA:
>
> | Shared LoRA path  | Routed LoRA path | FedAvg |
> | :-----|:----: |:----: |
> | 0.823 | 0.354 | 0.679 |
>
> Compared to FedAvg, the shared LoRA path exhibits substantially **higher** cross-client similarity, whereas the routed LoRA path shows much **lower** similarity. This indicates that the shared LoRA expert **captures knowledge that is common across samples and clients**, while the routed LoRA experts **model instance-level, high-variance local variations**. These results demonstrate that Fed-MoLE effectively disentangles global and local information, so that aggregating only the shared LoRA can reduce client drift while effectively integrating global knowledge to improve model generalization.
>
>
> > Q3. “1+1 < 1” phenomenon may be less critical.
>
> We agree that under non-IID data, PFL methods often achieve stronger local performance than GFL, and therefore they are meaningful baselines. However, the “1+1 < 1” phenomenon addresses a **more fundamental limitation** of FL rather than a baseline selection issue. It reveals that, under non-IID data, global aggregation can become counterproductive, yielding worse performance than standalone training. This degradation **cannot** be resolved by simply comparing or switching to PFL methods.
>
> Instead, understanding this failure mode provides a **principled guideline** for how to effectively balance global generalization with local adaptation. This is an issue that **both** GFL and PFL must ultimately confront. Fed-MoLE is explicitly designed based on this insight, rather than merely aiming to outperform PFL baselines.
>
>
>
> > Q4. Computational cost varies  $N$ and $K_r$.
>
>  We report the peak memory of Fed-MoLE under different $N$ and $K_r$ as below:
>
>
> | Setting |$K_r = 2, N=4$ | $K_r = 2, N=8$ | $K_r = 2, N=12$ |
> | :-----|:----: |:----: |:----: |
> | Memory |16.53 GB |16.75 GB| 17.13 GB |
>
> | Setting |$K_r = 2, N=8$ | $K_r = 4, N=8$ | $K_r = 8, N=8$ |
> | :-----|:----: |:----: |:----: |
> | Memory |16.75 GB|19.32 GB|23.58 GB|
>
> As the router **only** activates $K_r$ experts, this sparse structure ensures that increasing the number of routed LoRA experts $N$ does **not** significantly increase computational overhead. The **main factor** affecting computation is the number of activated experts $K_r$: as $K_r$ grows, the model becomes denser, resulting in additional gradient computations for more experts.
>
>
> > Q5. Fed-MoLE is applied only to the up_proj layer.
>
> We applied Fed-MoLE only to the up_proj layer because this is **sufficient** to capture instance-level variations. Applying MoLE to additional layers does **not** lead to further performance improvement **but** significantly increases computational overhead. Therefore, limiting MoLE to up_proj achieves the **best trade-off** between accuracy and efficiency.
>
>
> > Q6. Equation (6) contains a typographical error.
>
>  Thanks for pointing this out. Equation (6) has been revised to: $\boldsymbol{H}(\boldsymbol{X}) = f(\boldsymbol{X}; \boldsymbol{W}^0) + \boldsymbol{B}^{s,t} _ {k}\boldsymbol{A}^{s,t} _ {k}\boldsymbol{X} + \sum_{i=1}^N\rho _ i\boldsymbol{B} _ {k,i}^{r,t}\boldsymbol{A} _ {k,i}^{r,t}\boldsymbol{X}$

---

### Official Review · Reviewer_FUcn · 2025-10-31

**Soundness:** 3
**Presentation:** 3
**Contribution:** 3
**Rating:** 4
**Confidence:** 5

**Summary:**

The paper identifies a key issue in federated fine-tuning of multimodal large language models that under non-IID data, standard approaches can perform worse than local training alone, a phenomenon referred to as the “1+1<1” problem. To address this, the authors propose Fed-MoLE, which employs a mixture of LoRA experts with dynamic routing to capture client-specific variations and integrates an alternating disentanglement–alignment mechanism to unify cross-client knowledge. Experiments demonstrate that Fed-MoLE consistently outperforms both state-of-the-art federated learning methods and standalone local training, effectively overcoming the “1+1<1” barrier.

**Strengths:**

1.Highlights and rigorously addresses the underappreciated “1+1<1” failure mode in federated fine-tuning—ensuring FL methods at least beat local baselines.
2.Introduces a hybrid mixture-of-LoRA-experts design that balances personalization and collaboration, where the alternating disentanglement–alignment strategy effectively handles instance-level heterogeneity while promoting global consistency.
3.Strong experimental results on multimodal benchmarks showing consistent gains over both SOTA FL methods and local training under non-IID settings.

**Weaknesses:**

1.Regarding the “1+1<1” issue: most existing methods include ablation studies on the number of participating clients, typically concluding that performance improves as more clients join, which directly contradicts the authors’ claim of a “1+1<1” phenomenon.
2.The combination of Mixture-of-Experts (MoE) and LoRA has been widely explored in prior work. The proposed Fed-MoLE method appears to lack novelty, as it merely integrates MoE with LoRA, and the alternating disentanglement–alignment strategy is relatively simple and generic.
3.The paper claims that the alternating disentanglement–alignment mechanism reduces client drift and enhances global representation. However, it only explains that disentanglement captures shared patterns across samples, without clarifying how this mechanism specifically mitigates client drift.
4.In Table 1, which compares Fed-MoLE against state-of-the-art methods, its performance is nearly identical to, or even slightly worse than that of FedSA-LoRA. The authors do not provide any analysis or justification for this marginal (or negative) performance gap.

**Questions:**

1.In Figure 1, the authors compare the performance of the standalone (independent) approach against other FedLoRA methods. How was the test set split for this comparison? Can the fairness of this comparison be guaranteed? Please also explain Weakness 1.

---

> ### Author Response · Authors · 2025-11-21
> **Official Comment by Authors**
>
> > W1. “1+1<1” issue.
>
> We clarify this as follows:
> (1) Both Standalone and FedLoRA use exactly the **same** train/test split and follow an **identical** experimental protocol, ensuring that the performance differences are **not** caused by any unfairness in the evaluation.
>
> (2) Existing studies [1-3] on federated fine-tuning have also reported significant performance degradation under non-IID data. Therefore, the “1+1 < 1” phenomenon observed in our experiments is a **reliable and representative behaviour** that is **consistent** with their findings.
>
> (3) To further clarify this issue, we additionally conducted an experiment, where we randomly partitioned the AQUA dataset across clients under an IID setting and increased the number of clients K. The results show that the “1+1” phenomenon does **not** occur regardless of how many clients are involved. This confirms that the “1+1” phenomenon is from the non-IID data distribution across clients, **rather than** from the number of participating clients.
>
> | Method |K = 5 | K = 10 | K = 20 | K = 40 |
> | :-----|:----: |:----: |:----: |:----: |
> | Standalone | 70.57 |  68.35 | 65.42 | 63.11 |
> | FedAvg  | 71.76 | 69.98 | 67.92 | 64.40 |
>
> > W2. The method lacks novelty.
>
> The MoE+LoRA studies **emerged contemporaneously** with our work and primarily focus on addressing the compatibility issues of applying  **vanilla** MoE+LoRA in federated fine-tuning. Their goals and designs **differ substantially** from ours. The novelty of Fed-MoLE does **not** stem from the use of MoE+LoRA. Instead, our key contribution is **the first** to reveal the “1+1 < 1” phenomenon in federated fine-tuning of MLLMs, where naive FedLoRA can perform even worse than single-client training. Motivated by this, we introduce **a hybrid MoE+LoRA** architecture consisting of a shared LoRA expert that integrates cross-client knowledge and dynamically routed LoRA experts that capture instance-level, high-variance local patterns. Therefore, Fed-MoLE introduces **a new perspective and architectural formulation** rather than a straightforward application of MoE to federated LoRA fine-tuning.
>
> > W3. Disentanglement–alignment mechanism.
>
> We have analyzed the cross-client feature differences learned by the shared LoRA and routed LoRA experts. Before aggregation, we compute the pairwise cosine similarity between clients using the outputs of the two paths in the final tuned layer of Fed-MoLE, as well as the corresponding layer in FedAvg. The averaged results on Fed-VQA:
>
> | Shared LoRA path  | Routed LoRA path | FedAvg |
> | :-----|:----: |:----: |
> | 0.823 | 0.354 | 0.679 |
>
> Compared to FedAvg, the shared LoRA path exhibits substantially **higher** cross-client similarity, whereas the routed LoRA path shows much **lower** similarity. This indicates that the shared LoRA expert **captures knowledge that is common across samples and clients**, while the routed LoRA experts **model instance-level, high-variance local variations**. These results demonstrate that Fed-MoLE effectively disentangles global and local information, so that aggregating only the shared LoRA can reduce client drift while effectively integrating global knowledge to improve model generalization.
>
> > W4. Slight improvement in Table 1.
>
> (1) Prior FL studies [4-5] **typically** use global performance (Avg.) as the primary evaluation metric, as this **better** reflects the model's generalization capability. In Table 1, Fed-MoLE outperforms FedSA-LoRA in terms of global performance. Besides, we further conducted three independent trials using different random seeds on Fed-VQA, and reported the mean and standard deviation. Furthermore, we assessed the significance of the performance differences using the paired t-test. The results are shown below:
>
> | Method | Avg. | p-value |
> | :-----|:----: |:----: |
> | FedSA-LoRA | $69.62 \pm 0.52$ |  - |
> | Fed-MoLE  | $70.84 \pm 0.36$ | 0.011 |
>
> The experimental results indicate that Fed-MoLE shows a **significant improvement** over FedSA-LoRA in global performance (p-value < 0.05)
>
> (2) FedSA-LoRA indeed shows a strong advantage on individual clients, as its personalization strategy allows it to better fit each client’s data. However, it does **not** balance all clients well, as evidenced by its **lower** performance than the standalone model on AQUA and GQA. In contrast, Fed-MoLE is the **only** method that outperforms the standalone model across all clients, indicating that it enables all clients to benefit from FL, which aligns **better** with practical value.

---

> ### Author Response · Authors · 2025-11-21
> **Official Comment by Authors**
>
> > Q1. Experimental setup in Figure 1.
>
> In Figure 1, there are five clients corresponding to five different VQA datasets. Each dataset is randomly split into a training set and a test set, with the statistics shown in **Table 6**. We train the models on the training sets and the well-trained model is evaluated on the corresponding client’s local test set. We report the performance for each client separately, and Avg. denotes the average score across all clients. This setup is widely used in prior FL studies with non-IID data [4-5]. For both Standalone and FedLoRA, we use exactly the **same** train/test split and follow an **identical** experimental protocol, ensuring that the performance differences are not caused by any unfairness in the evaluation.
>
> **References**
>
> [1] Improving LoRA in Privacy-preserving Federated Learning. ICLR, 2024.
>
> [2] Selective Aggregation for Low-Rank Adaptation in Federated Learning. ICLR, 2025.
>
> [3] Federated Residual Low-Rank Adaptation of Large Language Models. ICLR, 2025.
>
> [4] FedFA: Federated Feature Augmentation. ICLR, 2023.
>
> [5] Rethinking Federated Learning with Domain Shift: A Prototype View. CVPR, 2023

---

### Official Review · Reviewer_Ad8a · 2025-11-07

**Soundness:** 3
**Presentation:** 3
**Contribution:** 2
**Rating:** 4
**Confidence:** 4

**Summary:**

The paper studies federated fine-tuning of multimodal large language models under non-IID conditions. The authors argue that the common “share a single LoRA adapter” approach can yield “1+1<1” in strongly non-IID settings, federated training may even underperform standalone fine-tuning on each client. They propose Fed-MoLE: at each target layer, introduce a mixture-of-experts structure with a shared LoRA plus multiple routed LoRA experts. Training alternates in a decoupled manner, first update routed experts and the router, then only update the shared LoRA; only the shared LoRA is aggregated. On Fed-VQA (5 clients) and Fed-Med (3 clients with task heterogeneity), the method reportedly outperforms Standalone and several FL baselines, with no extra communication cost and only modest VRAM overhead.

**Strengths:**

1. Clear and practically meaningful problem setup. The paper emphasizes a sensible baseline principle—federated fine-tuning should at least not underperform Standalone—surfacing an evaluation pitfall overlooked by many works. The “client drift / representation misalignment” diagnosis (variance and norm traces) is intuitive and useful.
2. Simple and effective method. Aggregate only the shared LoRA, leaving instance-level heterogeneity to sparse routed experts. Alternating freezing reduces gradient conflicts; the engineering recipe looks implementable (with pseudocode and practical notes).
3. Efficiency considerations. Communication and VRAM comparisons are laid out; communication matches standard LoRA-FL and VRAM grows only modestly.

**Weaknesses:**

1. Insufficient theoretical support. The paper does not clearly state modeling assumptions for the non-IID setting (e.g., bounds on gradient dissimilarity or a “shared subspace + client-specific shift” decomposition), making it difficult to delineate under what conditions “aggregating only the shared LoRA” is effective; the training dynamics lack convergence/stability guarantees—especially under federated settings with partial participation, asynchrony, and distribution shift—so error bounds and the radius of convergence remain unclear; the interaction between the shared LoRA and expert LoRA, as well as the impact of routing misassignment rates on optimization and generalization, is not quantitatively analyzed; moreover, the absence of generalization-error or sample-complexity bounds leaves the expected gains under task heterogeneity and data imbalance unexplained.

2. Notational issues. At least in Eq. (6), the routed expert term appears as B^r B^r X,which likely should be B^r A^r X under LoRA. 	Clarifications needed on design and evaluation details. The paper under-specifies the router’s architecture, its inputs, the regularization terms, and any perturbation strategies; moreover, it does not analyze the stability of Top-K sparse routing under distribution shift. The claim of “no extra communication cost” also requires stricter accounting—including the sizes of shared vs. local parameters, the actual transmitted bytes under sparse activation, and the impact of optional quantization/compression. As presented, the evidence is largely qualitative and based on single-round measurements.
3. Comparisons to personalized/clustered/expert-style FL are incomplete. The paper mainly compares with global/personalized LoRA-FL variants; however, MoE or clustering-style personalization is not new in FL. Missing strong baselines from this line dilutes the novelty claim.

**Questions:**

see weaknesses.

---

> ### Author Response · Authors · 2025-11-21
> **Official Comment by Authors**
>
> > W1. Insufficient theoretical support.
>
> (1) We have analyzed the cross-client feature differences learned by the shared LoRA and routed LoRA experts. Before aggregation, we compute the pairwise cosine similarity between clients using the outputs of the two paths in the final tuned layer of Fed-MoLE, as well as the corresponding layer in FedAvg. The averaged results on Fed-VQA:
>
> | Shared LoRA path  | Routed LoRA path | FedAvg |
> | :-----|:----: |:----: |
> | 0.823 | 0.354 | 0.679 |
>
> Compared to FedAvg, the shared LoRA path exhibits substantially **higher** cross-client similarity, whereas the routed LoRA path shows much **lower** similarity. This indicates that the shared LoRA expert **captures knowledge that is common across samples and clients**, while the routed LoRA experts **model instance-level, high-variance local variations**. These results demonstrate that Fed-MoLE effectively disentangles global and local information, so that aggregating only the shared LoRA can reduce client drift while effectively integrating global knowledge to improve model generalization.
>
> (2) Our empirical analysis in **Figure 4** further shows that **Fed-MoLE converges stably and achieves convergence comparable to standalone**.
>
> (3) We have added a generalization-bound analysis of our method in the revised manuscript. Please refer to **Appendix D** for details.
>
> > W2. Notational issues.
>
> (1) Thanks for pointing this out. Equation (6) has been revised to: $\boldsymbol{H}(\boldsymbol{X}) = f(\boldsymbol{X}; \boldsymbol{W}^0) + \boldsymbol{B}^{s,t} _ {k}\boldsymbol{A}^{s,t} _ {k}\boldsymbol{X} + \sum_{i=1}^N\rho _ i\boldsymbol{B} _ {k,i}^{r,t}\boldsymbol{A} _ {k,i}^{r,t}\boldsymbol{X}$
>
> (2) The router is **a lightweight MLP (one layer)** that maps $\boldsymbol{X}$ (i.e., the output of the previous layer) to a score vector $z$, which controls the activations of routed LoRA experts. No additional regularization terms or perturbation strategies are employed in our method.
>
> (3) We evaluate the stability of the Top-K routing by entropy. Specifically, we compute the average entropy over all routers on the training and test sets of AQUA. The results show that the router entropy is similar on both the training and test sets, indicating that the Top-K routing remains **stable** under distribution shift.
> | Method | training | test |
> | :-----|:----: |:----: |
> | Entropy | 0.486 | 0.521 |
>
> (4) Fed-MoLE's communication overhead = number of shared LoRA experts X size of LoRA. Since only one shared LoRA expert is used, the communication cost of Fed-MoLE is identical to that of FedLoRA, ensuring a fair comparison. The size of shared vs. local parameters in Fed-MoLE with LLaVA-1.5-7B is presented below.
>
> | shared parameters| local parameters|
> | :-----|:----: |
> | 226.8 MB | 206.3 MB |
>
>
>
> > W3. More baselines.
>
> We have compared our method against FedLEASE [1] in the revision (Tables 1 and 2). The results, as presented below, show that our method consistently outperforms FedLEASE on both Fed-VQA and Fed-Med.
>
>
> *Table 1: Fed-VQA*
> | Method      | AQUA   |GQA  | VizWiz  | Abstract   | COCO-QA | Avg. |
> |-------------|-------|-------|-------|--------|--------|--------|
> | FedLEASE    | 70.77 | 70.31 | 63.12 | 66.53 | 77.40| 69.71  |
> | **Fed-MoLE (Ours)** | 74.95 | 71.10 | 61.32 | 64.96 | 79.37 | **70.38** |
>
> *Table 2: Fed-Med*
> | Method      | VQA   | Report Generation  | Detection | Avg. |
> |-------------|-------|-------|-------|--------|
> | FedLEASE    | 64.50 | 239.71 | 43.37 | 115.86 |
> | **Fed-MoLE (Ours)** | 68.00 | 260.00 | 46.38 | **124.79** |
>
> [1] Adaptive LoRA Experts Allocation and Selection for Federated Fine-Tuning. NeurIPS, 2025.

---

### Author Response · Authors · 2025-11-21
**General Response**

Dear reviewers and meta-reviewers,

We appreciate all reviewers for their valuable comments and suggestions. We've revised our manuscript based on reviewers' comments as follows:

1. **For Reviewer Ad8a and n4EL**, we have revised the **Equation (6)**.

2. **For Reviewer Ad8a**, we have included a theoretical analysis of the **generalization bound** in **Appendix D**.

3. **For Reviewer Ad8a, FUcn and n4EL**, we have added an empirical analysis of our **disentanglement–alignment mechanism** in **Appendix B.5 Table 9**.

4. **For Reviewer Ad8a**, we have added the results of **FedLEASE** in **Tables 1 and 2**.

5. **For Reviewer 2zi3**, we have conducted an ablation study with different values of $r$ in **Appendix B.6 Table 10**.

The changes have been highlighted in **blue** in the revised paper. Please see below for our responses to each reviewer. If you have any further questions or suggestions, please feel free to share them on OpenReview.

---

### Meta-Review · Area_Chair_S9JP · 2026-01-06

**Summary:**

### Summary:

This paper studies federated fine-tuning of multimodal large language models (MLLMs) under non-IID data and highlights a “1+1<1” phenomenon, where naïve federated training with a single LoRA adapter can underperform standalone fine-tuning on each client. To address this, the authors propose Fed-MoLE, a Mixture-of-LoRA-Experts framework in which each target layer has (i) a shared LoRA expert aggregated across clients and (ii) dynamically routed local LoRA experts. Training alternates between updating routed experts + router and updating only the shared LoRA, and only the shared LoRA is globally aggregated. Experiments on Fed-VQA (5 clients) and Fed-Med (3 heterogeneous clients) show that Fed-MoLE outperforms standalone and several FL baselines.

### Strengths:

1.  **Relevant and clearly stated problem:** The paper highlights a highly relevant and practically important problem in federated fine-tuning: the "1+1<1" phenomenon, where collaboration can be detrimental under non-IID conditions.
2.  **Conceptually simple and implementable method.**
3.  **Solid empirical evaluation and additional analyses in rebuttal.**

### Weaknesses:

1.  **Limited Novelty (Reviewers FUcn, n4EL, 2zi3):** Reviewers consistently found the technical contribution to be somewhat limited. The use of MoE with LoRA in FL has already been widely explored in prior work.
2.  **Insufficient Depth of Theoretical/Formal Justification (Reviewer Ad8a):** The reviewer requested sufficient theoretical convergence justification for the proposed method.
3.  **Insufficient Experiments (Reviewer Ad8a, n4EL):** Reviewers noted that the paper does not 1) provide comparisons with MoE or clustering-style personalization FL methods, 2) a computational costs analysis.
4.  **“1+1 < 1” Issue (Reviewers FUcn, n4EL, 2zi3):** Reviewers acknowledged that the “1+1 < 1” phenomenon is interesting, but raised concerns regarding its causes. The paper lacks further investigation into the underlying reasons and does not provide a strong connection between the reported phenomenon and the proposed method.
5.  **Lack of Mechanistic Explanation (Reviewers n4EL, FUcn):** There is no in-depth clarification of why the routed LoRA models instance-level variations and the shared LoRA captures cross-sample shared features. There is also no clear explanation of how the alternating disentanglement–alignment mechanism mitigates client drift.
6.  **Unclear Experimental Settings (Reviewer FUcn)**
7.  **Design Choice for Applying Fed-MoLE Only to the `up_proj` Layer (Reviewer n4EL).**
8.  **Modest Gains Relative to Personalized FL Methods (Reviewers FUcn, 2zi3)** such as FedSA-LoRA.

### Decision:

The paper initially received consistent "Weak Rejection" scores (4, 4, 4, 4) from all reviewers. These initial assessments highlighted several critical concerns, including limited methodological novelty, insufficient in-depth analysis, and only marginal improvements when compared to state-of-the-art methods. Despite a commendable rebuttal from the authors, which successfully addressed some of the raised issues, fundamental concerns persist. The most significant issues remain the lack of novelty and insufficient in-depth analysis. Reviewers consistently found the technical contribution to be somewhat limited, noting that the use of MOE with LoRA in FL has already been widely explored in prior work. While the authors attempted to clarify that the novelty of Fed-MoLE does not solely stem from the MoE+LoRA combination itself, but rather from being the first to reveal the “1+1 < 1” phenomenon in federated fine-tuning of MLLMs, this explanation did not fully alleviate the concerns regarding the methodological contribution. Furthermore, regarding the mechanistic understanding of why the proposed method works and its theoretical underpinnings, the authors' attempts at providing analysis were not fully convincing. The added theoretical analysis is not specifically tailored to Fed-MoLE method. Considering these persistent limitations in methodological novelty, theoretical grounding, and the depth of mechanistic explanation, especially when evaluated against stronger submissions in my batch, I recommend rejection.

**Reviewer Concerns:**

1.	Limited Novelty (Reviewers FUcn, n4EL, 2zi3): Reviewers consistently found the technical contribution to be somewhat limited, noting that the use of MoE with LoRA in FL has already been widely explored in prior work. During the rebuttal, the authors clarified that the novelty of Fed-MoLE lies not in the MoE+LoRA combination, but in being the first to reveal the “1+1 < 1” phenomenon in federated fine-tuning of MLLMs. Nevertheless,  this explanation did not fully alleviate the concerns regarding the methodological contribution.
2.	Insufficient Depth of Theoretical/Formal Justification (Reviewer Ad8a): Reviewers requested sufficient theoretical convergence justification for the proposed method. In the rebuttal, the authors provided a formal generalization bound in Appendix D. However, this analysis is not tailored to Fed-MoLE: the roles of the shared LoRA, routed LoRA experts, and the alternating training strategies are not considered at all. Therefore, the presented bound cannot be regarded as a formal proof of convergence for the proposed method.
3.	Insufficient experiments (Reviewer Ad8a, n4EL): Reviewers noted the lack of (1) comparisons with MoE or clustering-style personalization FL methods, and (2) computational cost analysis. These concerns were successfully addressed in the rebuttal by adding extra experiments and analysis.
4.	 “1+1 < 1” Issue (Reviewers FUcn, n4EL, 2zi3):  Reviewers found the “1+1 < 1” phenomenon interesting but raised concerns about its underlying causes. The paper initially lacked further investigation and did not strongly connect the phenomenon to the proposed method. Moreover, previous works generally find that FL performance improves with more clients, which contradicts the claimed phenomenon. The authors provided empirically intuitive explanations and clarifications regarding why this occurs, partially addressing the concerns (no in-depth analysis is provided).
5.	Lack of Mechanistic Explanation (Reviewers n4EL, FUcn): Reviewers noted the absence of in-depth clarification of why routed LoRA models capture instance-level variations and shared LoRA captures cross-sample shared features. Additionally, there was no clear explanation of how the alternating disentanglement–alignment mechanism mitigates client drift. The authors attempted to address this by analyzing cross-client feature differences learned by the shared and routed LoRA experts, which partially addresses the concern, though the underlying reasons remain unexplained.
6.	Unclear Experimental Settings (Reviewer FUcn), which was fully addressed during rebuttal.
7.	Design Choice for Applying Fed-MoLE Only to the up_proj Layer (Reviewer n4EL):  The reasoning for applying Fed-MoLE only to the up_proj layer was only partially addressed. The authors explained that this layer is sufficient to capture instance-level variations, and applying MoLE to additional layers does not lead to further performance improvement but significantly increases computational overhead.
8.	Modest Gains Relative to Personalized FL Methods (Reviewers FUcn, 2zi3): Reviewers noted that the improvements over strong personalized FL baselines such as FedSA-LoRA are relatively modest. The authors clarified that Fed-MoLE outperforms FedSA-LoRA in global performance and on AQUA and GQA.

**Reviewer Scores:**

Reviewer Ad8a (Rating: 4 -> keep the same), Reviewer FUcn (Rating: 4 ->  may be  slightly improved or keep the same), Reviewer n4EL (Rating: 4 -> maybe keep the same or slightly improved), Reviewer 2zi3 (Rating: 4 -> keep the same)

---

### Decision · Program_Chairs · 2026-01-26

Reject